# Modeling cancer genomic data in yeast reveals selection against ATM function during tumorigenesis

**Marcel Hohl**[1], **Aditya Mojumdar**[2], **Sarem Hailemariam**[3], **Vitaly Kuryavyi**[4], **Fiorella Ghisays**[1], **Kyle Sorenson**[2], **Matthew Chang**[5], **Barry S. Taylor**[5], **Dinshaw J. Patel**[4], **Peter M. Burgers**[3], **Jennifer A. Cobb**[2], **John H. J. Petrini**[1] *

**1** Molecular Biology Program, Memorial Sloan-Kettering Cancer Center, New York, New York, United States of America, **2** Departments of Biochemistry & Molecular Biology and Oncology, Robson DNA Science Centre, Arnie Charbonneau Cancer Institute, Cumming School of Medicine; University of Calgary, Calgary, Canada, **3** Department of Biochemistry and Molecular Biophysics, Washington University School of Medicine, St. Louis, Missouri, Untied States of America, **4** Structural Biology Program, Memorial Sloan-Kettering Cancer Center, New York, New York, United States of America, **5** Marie-Josée and Henry R. Kravis Center for Molecular Oncology, Memorial Sloan Kettering Cancer Center, New York, NY, USA; Human Oncology and Pathogenesis Program, Memorial Sloan Kettering Cancer Center, New York, New York, United States of America

* petrinij@mskcc.org

**Data Availability Statement:** All relevant data are within the manuscript and its Supporting Information files.

## Abstract

The DNA damage response (DDR) comprises multiple functions that collectively preserve genomic integrity and suppress tumorigenesis. The Mre11 complex and ATM govern a major axis of the DDR and several lines of evidence implicate that axis in tumor suppression. Components of the Mre11 complex are mutated in approximately five percent of human cancers. Inherited mutations of complex members cause severe chromosome instability syndromes, such as Nijmegen Breakage Syndrome, which is associated with strong predisposition to malignancy. And in mice, Mre11 complex mutations are markedly more susceptible to oncogene- induced carcinogenesis. The complex is integral to all modes of DNA double strand break (DSB) repair and is required for the activation of ATM to effect DNA damage signaling. To understand which functions of the Mre11 complex are important for tumor suppression, we undertook mining of cancer genomic data from the clinical sequencing program at Memorial Sloan Kettering Cancer Center, which includes the Mre11 complex among the 468 genes assessed. Twenty five mutations in *MRE11* and *RAD50* were modeled in *S. cerevisiae* and *in vitro*. The mutations were chosen based on recurrence and conservation between human and yeast. We found that a significant fraction of tumor-borne *RAD50* and *MRE11* mutations exhibited separation of function phenotypes wherein Tel1/ATM activation was severely impaired while DNA repair functions were mildly or not affected. At the molecular level, the gene products of *RAD50* mutations exhibited defects in ATP binding and hydrolysis. The data reflect the importance of Rad50 ATPase activity for Tel1/ATM activation and suggest that inactivation of ATM signaling confers an advantage to burgeoning tumor cells.

**Funding:** This work was supported by a Stand Up To Cancer-Ovarian Cancer Research Fund-Ovarian Cancer National Alliance-National Ovarian Cancer Coalition Dream Team Translational Cancer Research Grant (SU2C-AACR-DT16-15), GM56888, P01 CA087497 and MSK Cancer Center Core Grant P30 CA008748 (J.H.J.P.), the Maloris Foundation (D.J.P.), Anna Fuller Fund and the Josie Robertson Foundation (B.S.T), GM118129 (P.M. B.) and CIHR MOP-82736; MOP-137062 and NSERC 418122 (J.A.C.). The funders had no role in study design, data collection and analysis, decision to publish, or preparation of the manuscript.

**Competing interests:** The authors have declared that no competing interests exist.

## Author summary

A complex network of functions is required for suppressing tumorigenesis. These include processes that regulate cell growth and differentiation, processes that repair damage to DNA and thereby prevent cancer promoting mutations and signaling pathways that lead to growth arrest and programmed cell death. The Mre11 complex influences both signaling and DNA repair. To understand its role in tumor suppression, we characterized mutations affecting members of the Mre11 complex that were uncovered through cancer genomic analyses. The data reveal that the signaling functions of the Mre11 complex are important for tumor suppression to a greater degree than its role in DNA repair.

## Introduction

The Mre11 complex, consisting of dimers of Mre11, Rad50, and Xrs2 in budding yeast (or Nbs1 in fission yeast and other eukaryotes) plays a central role in the DNA damage response (DDR). It is a primary sensor of DNA double strand breaks (DSBs) and thus situated activates the transducing kinase Tel1/ATM. In addition, the complex is required for both homology directed DNA repair (HDR) and non-homologous end-joining (NHEJ) [1,2].

The Mre11 complex has an elongated structure characteristic of the structural maintenance of chromosomes (SMC) protein family [3,4], comprising dimers of each of its three components. The globular domain is the site of DNA binding and houses the complex's enzymatic activities; Rad50 ATPase and the Mre11 nuclease. The Walker A and B ATPase motifs of Rad50 are separated by an extended coiled coil domain that folds back on itself in an antiparallel fashion. Dimeric Rad50 is a bipartite ATPase in which two ATP molecules are coordinated in trans such that the Walker A of one protomer and the Walker B of the other engage an ATP molecule. ATP-mediated dimerization of the globular domain and its disengagement upon ATP-hydrolysis induces large-scale structural transitions of the complex from a closed (ATP bound) to an open (ATP hydrolyzed) state. Available evidence suggests that those conformational states mediate distinct Mre11 complex functions [2,5].

Perhaps consistent with this interpretation, the complex's role in activating Tel1/ATM can be genetically separated from its DSB repair functions [6–8], suggesting that their underlying mechanisms are distinct. For example, $Rad50^{L1237F}$, a mutation found in an urothelial tumor, ($rad50$-$L1240F$ in yeast) selectively impairs Tel1/ATM activation [9]. The altered residue is located in the Rad50 ATP binding domain, supporting the idea that Rad50 ATPase activity and the structural transitions attendant to ATP binding and hydrolysis are integral to Tel1/ATM activation [2,5]. In addition, mutations that impair the association of the Mre11 complex with Tel1/ATM or Mre11 complex DNA binding can also selectively impair Tel1/ATM activation [10–12].

The identification of Mre11 complex tumor-borne alleles and modeling of their consequences in mice and yeast has been invaluable for understanding the Mre11 complex's roles in the DDR [13]. For example, phenotypic analysis of cells established from affected persons revealed the role of the Mre11 complex in activating ATM mediated DNA damage signaling.

We have shown in mouse models that the Mre11 complex is critical for suppressing oncogene induced breast carcinogenesis [14]. Having identified the $Rad50^{L1237F}$ mutation as the underlying cause of an extraordinary response to chemotherapy, likely due to its selective inability to activate Tel1/ATM, we reasoned that cancer genomic data could offer a rich source for understanding Mre11 complex functions. The principle being that the development of

malignancy could be viewed as a genetic screen for mutations affecting the processes that suppress malignancy. Herein, we modeled twenty-five *RAD50* and *MRE11* tumor alleles in yeast and *in vitro* in an attempt to shed light on the tumor suppressive function(s) of the complex. We prioritized recurrent mutations that affected conserved residues in *MRE11* and *RAD50*. We found that ten of the modeled alleles, including one that occurred in 16 distinct cancers, were severely impaired in Tel1/ATM activation. Collectively these data suggest that selection against ATM activation occurs during the progression of malignancy.

## Results

We identified the *Rad50^{L1237F}* mutation in an urothelial tumor that exhibited an extraordinary response to an otherwise ineffective combination of irinotecan and Chk1 inhibition. Modeling the mutation in yeast (*rad50-L1240F*) and mouse embryonic fibroblasts (MEFs) revealed that *rad50-L1240F* is a **S**eparation **O**f **F**unction (SOF) mutation that blocks Tel1/ATM activation while affecting DSB repair to lesser extents. The therapeutic efficacy of irinotecan in this context was interpreted to reflect the coincident defects in ATM activation—through *Rad50^{L1237F}*—and Chk1 activity—via inhibition [9]. As a result of this finding, *RAD50*, *NBS1*, and *MRE11* were added to the MSKCC IMPACT (**I**ntegrated **M**utation **P**rofiling of **A**ctionable **C**ancer **T**argets) gene list [15]. Mre11 complex genes have since been resequenced in over 40,000 tumors with mutations or copy number alterations found in roughly five percent of solid tumors [16].

We reasoned that additional functional analyses of Mre11 complex mutations arising in human cancer could provide insight regarding the mechanism(s) of Mre11 complex function including its role in tumor suppression. In total twenty-five mutations in *RAD50* and *MRE11* were modeled in *S. cerevisiae* (S1 and S2 Tables). The mutations modeled were chosen on the basis of conservation (thus encompassing mostly residues within Walker A and B domain; Fig 1B), recurrence (number of tumors), and the allele frequency (percentage of sequence reads of a given tumor that contain this mutation) observed. *NBS1* was not assessed in this study. The phenotypic parameters analyzed included Tel1/ATM activation, DNA repair, telomere length and the ability to produce viable spores.

As an overview, of the twenty-five mutations modeled, ten were found to be inconsequential. Three *rad50* (*R1217C, E1235K, and D1241N*) and two *mre11* alleles *(A173V, D370Y)*, each of which appeared to be heterozygous in their respective tumor, conferred defects in DSB repair when modeled in yeast (S1 Table).

Of the remaining fifteen, ten exhibited a separation of function phenotype similar to that of the *Rad50^{L1237F}* mutation: Tel1/ATM activation was severely impaired while DSB repair functions were largely intact. The SOF mutations fell into two classes. While both classes exhibited severely impaired Tel1/ATM activation, one class exemplified by *mre11-E38K* and *rad50-L1240F* depend on Tel1 protein for their DNA repair functions. The separations of DNA repair and Tel1/ATM activation was less pronounced in the first class; both *mre11-E38K rad50-L1240F* exhibited partial defects in DNA repair. *Rad50-D67N*/Y exemplified the second class. Their DNA repair functions are largely intact and independent of Tel1 protein. The mutations examined and the experimental approaches that led to these conclusions are described below.

Rad50-D69, which lies within the Walker A domain was mutated in sixteen tumors, either to asparagine (D69N; nine tumors), tyrosine (D69Y; six tumors) or glycine (D69G; one tumor) (Fig 1A and 1B and S1 Table). Within the Walker B domain, R1214C/H (five tumors), E1232K (two tumors), L1237F/V (two additional tumors) and R1256C/H (seven tumors) were

**A.**

| # | Cancer Type | Human | Yeast |
|---|---|---|---|
| 16 | Uterine Endometrioid Carcinoma (5)<br>Colorectal Adenocarcinoma (2)<br>Bladder Urothelial Carcinoma (1)<br>Myelodysplastic Syndromes (1) | D69N | D67N |
| | Lung Adenocarcinoma (5)<br>Acinic Cell Carcinoma (1) | D69Y | D67Y |
| | Lung Adenocarcinoma (1) | D69G | N/A |
| 5 | Colorectal Adenocarcinoma (2)<br>Cutaneous Squamous Cell Carc.(1)<br>Endometrial Cancer (1) | R1214C | R1217C |
| | Uterine Endometrioid Carcinoma (1) | R1214H | N/A |
| 2 | Lung Adenocarcinoma (1)<br>Lung Squamous Cell Carcinoma (1) | E1232K | E1235K |
| 3 | Small Cell Cancer Ureter (1*)<br>Cutaneous Melanoma (1) | L1237F | L1240F |
| | Breast Invasive Lobular Carcinoma (1) | L1237V | L1240V |
| 7 | Uterine Endometrioid Carcinoma (1)<br>Bladder Urothelial Carcinoma (2)<br>Signet Ring Cell Appendix (1)<br>Colorectal Adenocarcinoma (1) | R1256C | R1259C |
| | Uterine Carcinosarcoma (1)<br>Uterine Endometrioid Carcinoma (1) | R1256H | N/A |
| 4 | Ovarian Cancer (1)<br>Small Cell Lung Cancer (1) | E42K | E38K |
| | Uterine Endometrioid Carcinoma (1) | E42D | N/A |
| | Uterine Endometrioid Carcinoma (1) | E42A | N/A |

*RAD50* spans the first six row groups; *MRE11* spans the last row group.

**B.**

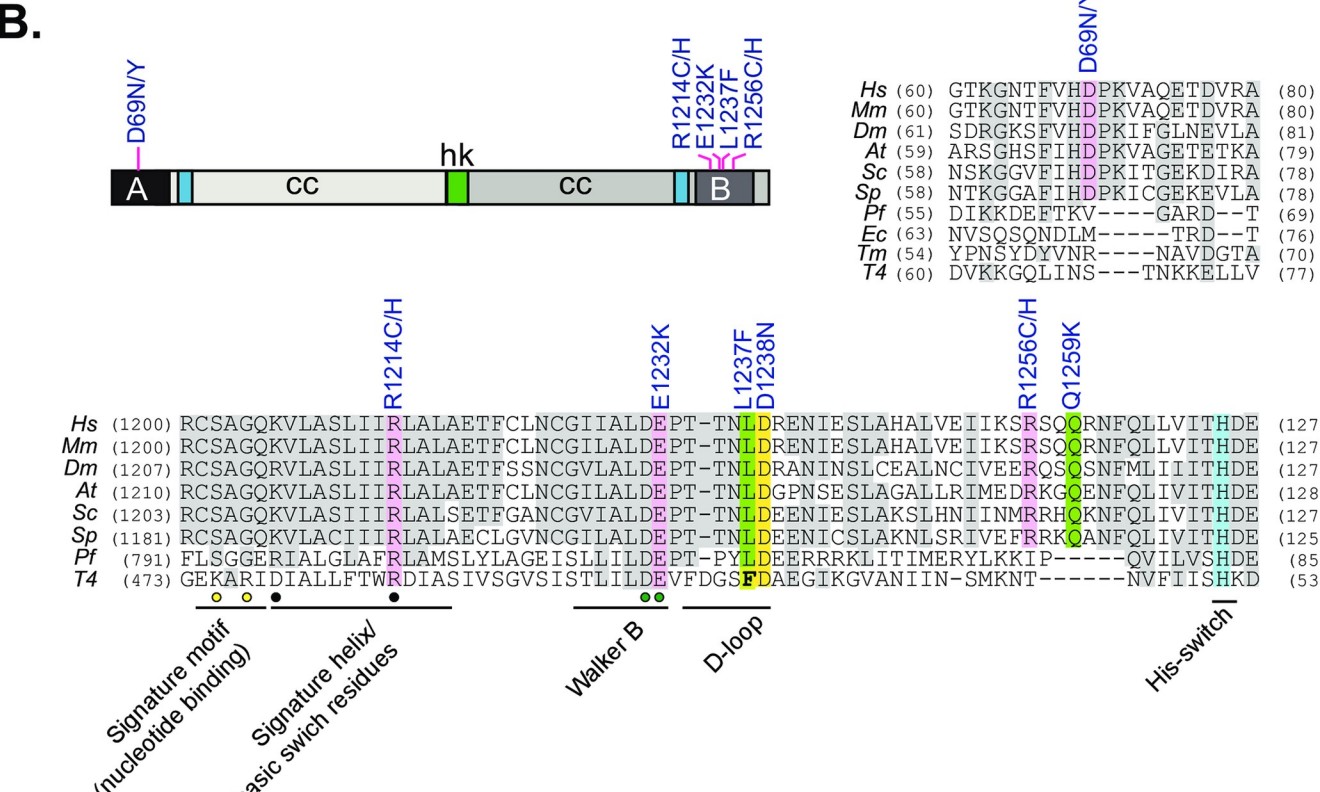

**Fig 1. *Rad50* and *Mre11* alleles modeled in yeast.** (A) The table lists the Rad50 and Mre11 alleles, cancer type (with number of instances denoted in brackets), and the corresponding yeast residues of the modeled alleles. Some alleles were not assessed (N/A) in this study. An extended version of above table including allele frequencies, number of mutations present in tumor sample and sequencing database source is available in S1 Table. *rad50-L1240F* was previously modeled based on an *RAD50^L1237F^* outlier patient (denoted by *) with an extraordinary response to chemotherapy [9] (B) Rad50 primary protein structure with the modeled mutations and relevant Rad50 domains are denoted (abbreviated A, B, cc, hk). The mutations modeled in this study are highlighted in color and conserved residues in grey in the multiple sequence alignment of Rad50 proteins. The *RAD50^D1238N^* and *RAD50^Q1259K^* alleles previously modeled in yeast [9] are also indicated. Known motifs in Walker B domain are denoted. Circles denote residues involved in specific binding of phosphates (yellow) and magnesium (green), respectively [40]. Black circles indicate basic switch residues mutated in previous studies [5,17]. Hs, *Homo sapiens*; Mm, *Mus musculus*; Dm, *Drosophila melanogaster*; At, *Arabidopsis thaliana*; Sc, *Saccharomyces cerevisiae*, Sp, *Schizosaccharomyces pombe*, Pf, *Pyrococcus furiosus*; Ec, *Escherichia coli*; Tm, *Thermotoga maritime*; T4, *Bacteriophage T4*.

identified. Finally, three *MRE11* SOF mutations were identified (E38K, D127N, R390C), but we only included E38K in further analyses since it exhibited a pronounced SOF phenotype.

## DSB Repair Functions In Tumor Modeled Mre11 Complex Mutants

Yeast *RAD50* and *MRE11* mutants corresponding to tumor borne alleles were integrated into a diploid yeast strain at their respective chromosomal loci, and haploid spores were derived by tetrad dissection.

Mre11 complex-mediated homologous recombination functions were inferred from cell survival in presence of methyl methanesulfonate (MMS), camptothecin (CPT) or hydroxyurea (HU) (Fig 2A). In Mec1-proficient cells, loss of Tel1 (*tel1Δ*) results in only mild sensitivity to the various clastogens, evident only at higher doses (Fig 2A, bottom lane). *rad50-R1217C* and *rad50-E1235K* phenocopied *rad50Δ*, and were inviable upon exposure to all three clastogens (see Fig 1A for numbering of human and yeast alleles), consistent with previous studies [5,17].

In contrast, the survival of *rad50-D67N*, *rad50-D67Y*, *rad50-R1259C*, *rad50-L1240F*, and *mre11-E38K* upon clastogen exposure was comparable to *WT*. *rad50-L1240F* and *mre11-E38K* were distinct in that they exhibited modest sensitivity at the highest dose (40 μM) of CPT, similar to the sensitivity observed in *tel1Δ* and *tel1KD* but still approximately 100-fold more resistant than *mre11-H125N* or *rad50-K81I*, which are defective in the removal of topoisomerase I and Spo11 adducts [18,19] (Fig 2A). These data indicate that these recurrent mutations do not compromise DSB repair by homologous recombination.

The Mre11 complex also promotes NHEJ. To assess that DSB repair mechanism in the modeled mutants, we created yeast strains in which a single DSB induced at the *MAT* locus must be repaired via NHEJ due to the lack of a homologous donor template [20]. *WT* cells plated on galactose-containing media to induce HO endonuclease expression exhibit 0.16% survival. The rate is at least 40-fold lower in *mre11Δ* and 8-fold lower in *rad50-K40A*, the protein product of which is defective in ATP binding [21,22]. The survival of the modeled mutants was indistinguishable from *WT* (Fig 2B), indicating that NHEJ is intact in those strains.

Colonies that survive chronic induction of the HO endonuclease do so because they have inactivated the HO site through Pol4-dependent addition (+CA; predominant in *WT* cells) or Pol2-dependent loss (ΔACA or ΔCA; predominant in *rad50Δ*) [20,23]. Although survival was unaffected, sequencing of the HO junctions revealed that in all the modeled mutants, survivors predominantly exhibited one to four nucleotide deletions, in contrast to the small insertions frequently seen in *WT* (Fig 2C). This difference in imprecise NHEJ-junctions between *WT* and mutants might reflect differences in recruitment of aforementioned polymerases or of other factors that influence DSB end processing.

In meiosis, the Mre11 complex is required at two distinct steps. First, the complex must be present for Spo11 to induce the DSBs that initiate meiotic recombination, and second, the complex is required for removal of Spo11 from the DSB end [19,24,25]. Loss of either function blocks the formation of viable spores. Diploids homozygous for the modeled mutations were

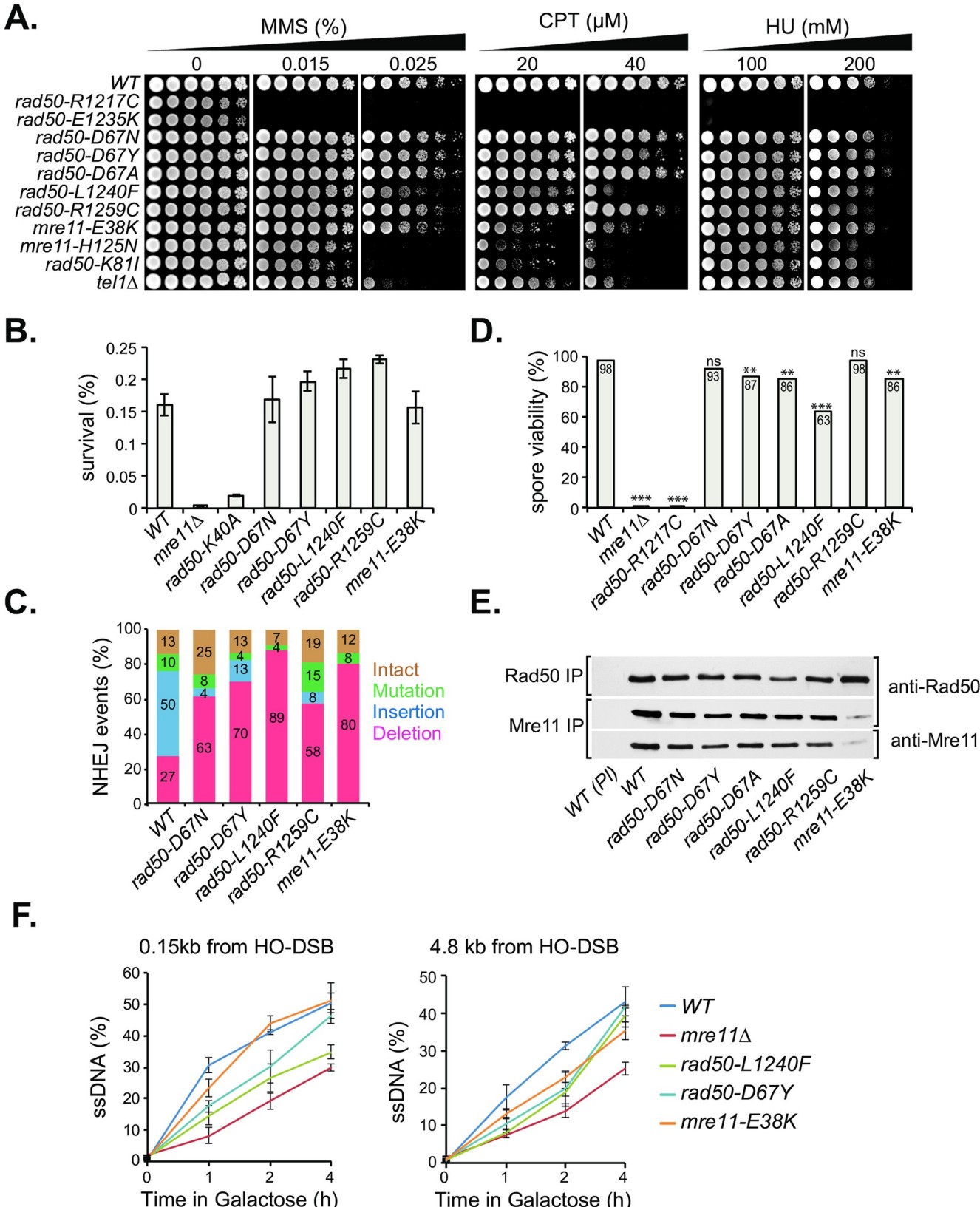

**Fig 2. DSB Repair Functions in Modeled Mre11 Complex Mutants.** (A) Clastogen sensitivities of modeled alleles. *Wild type* (*WT*), *mre11-H125N* (*mre11* nuclease dead allele), *rad50-K81I* (*rad50S*) and *tel1Δ* were included for comparison (B) Cell survival upon chronic HO-induction at the *MAT* locus. (C) The type and percentages of imprecise NHEJ events. (D) Spore viability assessed by tetrad dissection of at least 20 tetrads. Asterisks indicate a significant difference from WT (Fisher's exact test; **p<0.01, *** p<0.001). (E) Mre11 complex integrity in *wild-type* (*WT*) and modeled mutants assessed by co-immunoprecipitation and western blot with Rad50 or Mre11 antisera. Preimmune antibodies (PI) were included as negative controls. IP, immunoprecipitation. (F) Q-PCR based resection assay. Cells were cultures for 0, 1, 2 and 4 hours to induce DSB formation and resection was assessed 0.15 kb and 4.8 kb from the DSB by qPCR with *Rsa*I-digested genomic DNA. Error bars denote standard deviation from three experiments.

sporulated and spore viability was determined by tetrad dissection. As seen in the response to MMS, *rad50-R1217C* phenocopied Mre11 complex deficiency, with < 0.01% tetrads formed and < 1% spore viability. Spore viability was similar to *WT* (98%) in each of the other mutants examined (Fig 2D). These data indicate that both Spo11-mediated DSB formation and the Mre11 complex-dependent cleavage activity were unaffected in these mutants. An exception was seen in *rad50-L1240F*, which exhibited only 63% viable spores (Fig 2D).

Mre11 complex assembly was generally unaffected in all mutants tested, with the exception of *mre11-E38K*, in which Mre11-E38K protein levels were markedly reduced. The Mre11-Rad50 interaction was also intact in all mutants, based on comparable levels of Rad50 and/or Mre11 proteins recovered in Rad50 and Mre11 immunoprecipitations (Fig 2E). That *mre11-E38K* exhibited only a partial defect in DNA repair indicates that Mre11 complex levels are not limiting for the DSB repair functions tested, consistent with observations from previous studies [8,26,27].

Finally, additional physical assays were carried out to more fully examine DNA repair capacity of the SOF mutants. Mating type switching, as assayed by Southern blotting of HO-expressing *rad50-D67Y* and *rad50-L1240F* was indistinguishable from wild type (S1A Fig) [25,28]. We asked whether the partial CPT sensitivity and sporulation defect observed for *rad50-L1240F* might be due to an effect on Mre11 nuclease activity. Using a Q-PCR based assay for resection of an HO DSB [29], we noted reduced DSB resection in *rad50-L1240F* and to a lesser extent *rad50-D67Y* (Fig 2F). However, Mre11 nuclease activity is minimally affected, since in contrast to *mre11* nuclease dead alleles (*mre11-3*, *mre11-H125N*) [30,31], *rad50-L1240F*, *rad50-D67Y*, and *mre11-E38K* exhibit only mild synergy with *exo1Δ* or *sgs1Δ* with respect to CPT or MMS-sensitivity (S1B Fig). These data clearly demonstrate that the Mre11 nuclease is largely intact or only partially affected in these SOF mutants, especially in comparison to the profound decrement in Tel1 activation observed in those strains.

## Separation of function

In addition to its roles in DSB repair, the Mre11 complex is required for the activation of the Tel1/ATM axis of DDR signaling and cell cycle checkpoints [1]. We have shown that the DNA repair and DDR signaling functions of the complex are genetically separable [7–9]. In the *mec1Δ sae2Δ* context, Mec1-deficiency is suppressed in a Tel1- and Mre11 complex- dependent manner [32]. To assess whether Tel1 activation was affected in the modeled mutants, we crossed the mutants into a *mec1Δ sae2Δ* strain and assessed MMS sensitivity and Rad53 activation, both of which depend on Tel1 activity in that context.

In *WT* cells, *sae2Δ* suppressed *mec1Δ* MMS-sensitivity by over 500-fold at 0.006% MMS. *mec1Δ* suppression by *sae2Δ* was also evident in *rad50-D67N* and *rad50-D67Y*, but was much less pronounced than in *WT* cells (approximately 125-fold). No *mec1Δ* suppression by *sae2Δ* was observed in *rad50-L1240F*, *rad50-R1259C* and *mre11-E38K* mutants (Fig 3A).

Tel1 activation can also be queried via DNA damage induced phosphorylation of Rad53. The same *mec1Δ* and *mec1Δ sae2Δ* mutant strains were treated with MMS and Rad53 phosphorylation, as inferred from the appearance of slower migrating bands upon western blotting,

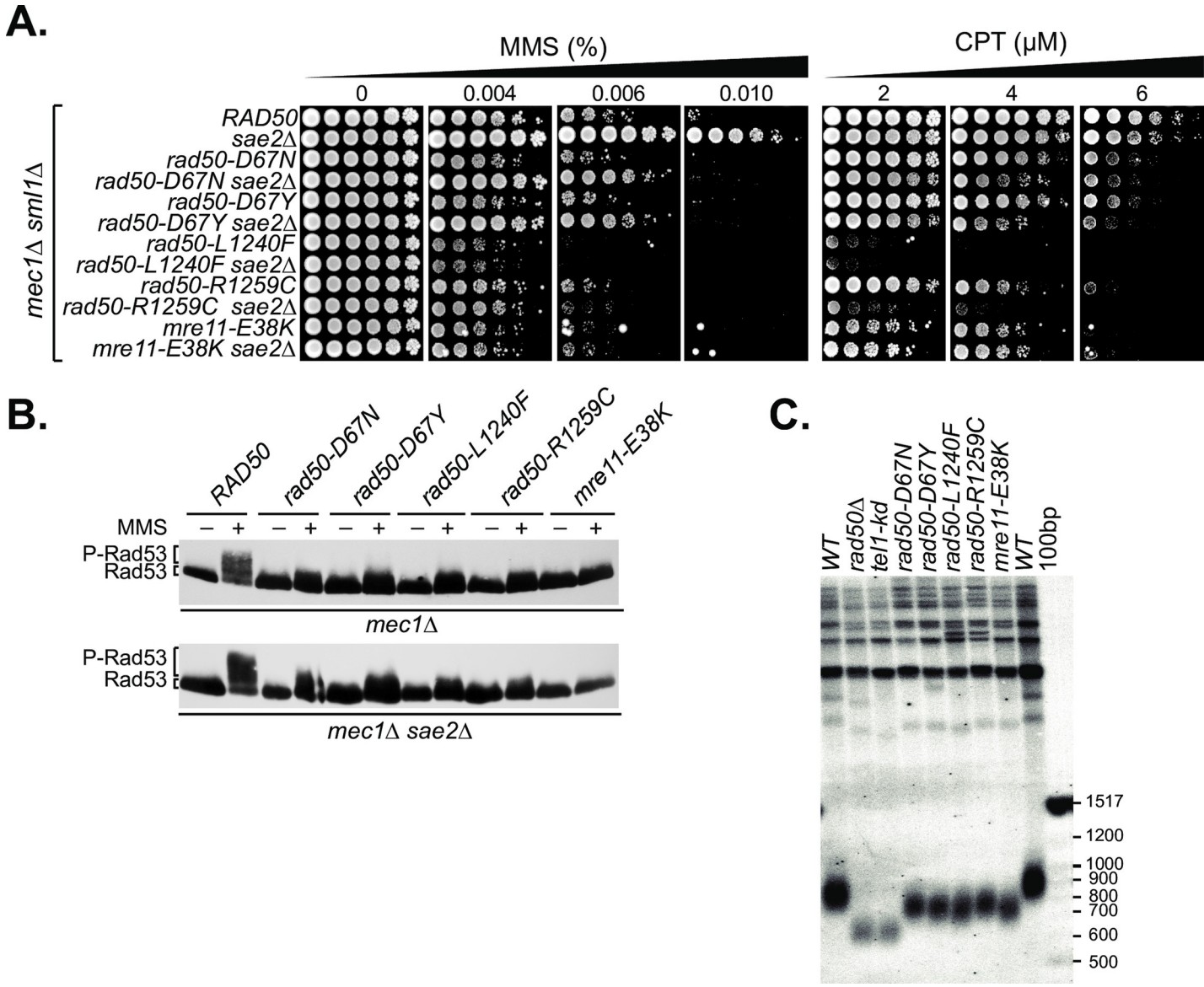

**Fig 3. Modeled mutants compromise the Mre11 complex-Tel1 dependent DNA damage response.** (A) MMS- and CPT- Survival of modeled Mre11 complex alleles in Mec1-and Mec1-Sae2-deficient background. All strains contained also *sml1Δ* to support viability in absence of Mec1. (B) Tel1-dependent Rad53 phosphorylation in *mec1Δ* and *mec1Δ sae2Δ* cells after MMS treatment (+) assessed by western blotting with anti-Flag-Rad53. Migration levels of the non-phosphorylated form (Rad53) and the phosphorylated form (P-Rad53) are indicated. (C) Telomere southern blot with *Pst*I-digested genomic DNA. The 100 bp DNA ladder (NEB) was detected by the telo-probe and size markers are given (measured in base pairs; bp).

was assessed. Whereas MMS treatment induced Rad53 phosphorylation in *mec1Δ sae2Δ* and to a lesser extent in *mec1Δ* cells, minimal to no phosphorylation was observed in *rad50-D67N/Y*, *rad50-R1259C*, *mre11-E38K* and *rad50-L1240F* cells (Fig 3B and S2 Fig). These data clearly show that Tel1 activation is impaired in *rad50-D67N/Y*, *rad50-R1259C*, *mre11-E38K* as well as in *rad50-L1240F* cells, while the DSB repair functions of those mutants were less severely affected. The defect in Tel1 activation was not complete, as telomere shortening, which is observed in *tel1Δ*, *tel1-kd*, and *rad50Δ*, was less severe in these mutants (Fig 3C). Given the severe Tel1/ATM activation defects observed in other experimental contexts, it appears that low levels of Tel1 activity are sufficient for telomere length regulation.

As with *rad50^hook* mutations [8], the *rad50-L1240F* mutation can be suppressed by mutations in the coiled coil domain (S343P, A1079T), as well as also by *rad50-L1240F* proximal and distal mutations (I23V, S1247N) within the globular domain (S3 Fig).

In summary, our study reveals that tumor borne alleles (*rad50-D67N/Y*, *rad50-R1259C*, *mre11-E38K*, and *rad50-L1240F*) found in 25 distinct tumors (Fig 1A) exhibit separation of function phenotypes specifically impairing Tel1 activation, while only partially affecting DSB repair.

## Molecular Phenotypes of *rad50* SOF Mutations

Mechanisms that could account for the Tel1 activation defects observed in the *rad50* SOF mutations include failure to recruit Tel1 to sites of DNA damage, failure of the mutant gene products to bind DNA, or failure to interact with Tel1. To investigate if impaired Tel1 signaling activity is due to reduced Mre11 complex or Tel1 DSB recruitment, we measured Xrs2-HA and Tel1-HA association to a HO-DSB at the *MAT* locus by chromatin immuno-precipitation (ChIP) and quantitative PCR (qPCR) with primers 0.6 kb and 1.6 kb from the DSB (Fig 4A).

The data revealed diverse effects on this outcome. Xrs2-HA enrichment (a surrogate for Mre11 complex recruitment) at the HO site three hours after DSB-induction was similar to *WT* (11.8-fold) in *rad50-D67N* (7.6-fold), *rad50-D67Y* (8.1-fold), *mre11-E38K* (10.6-fold) and *rad50-L1240F-S343P* (14.2-fold). However, DSB association was strongly reduced in *rad50-R1259C* to 1.7 percent enrichment and significantly increased in *rad50-L1240F* (24.6-fold). *rad50-K81I (rad50S)*, which chronically activates Tel1 even in the absence of DNA damage [32] exhibited markedly increased DSB association (49.1 fold enrichment) as previously shown [33] (Fig 4A, left panel).

The presence of Tel1 at the DSB did not follow the same trend as Mre11 complex recruitment to the DSB. As with Mre11 complex recruitment, DSB enrichment was similar in *WT* (3.2-fold), *mre11-E38K* (3.4 fold), and *rad50-L1240F-S343P* (4.8-fold), and increased in *rad50-L1240F* (8.1-fold) and *rad50-K81I* (12.6-fold). However, Tel1 DSB association was reduced or absent in *rad50-D67N*, *rad50-D67Y* and *rad50-R1259C*. In this regard, the mutations phenocopied *rad50-K40A*, the gene product of which does not bind ATP or DNA [21,34] (Fig 4A, right panel). These data indicate that impaired Tel1 activation in the SOF mutants is not solely due to defects in Tel1-recruitment, and suggests that other molecular defects such as defective ATP binding or hydrolysis may underlie this phenotype.

The levels of Rad50-L1240F and Mre11-E38K proteins are reduced *in vivo* (Fig 2E), and the corresponding strains exhibit DSB repair defects at 37˚C (S4 Fig). Recent studies suggest that Tel1 has a structural role in stabilizing Mre11 complex DSB association that is independent of its kinase activity [10,35,36]. Accordingly, we asked whether the apparently meta-stable Mre11 complexes that contain those gene products were dependent on Tel1 protein for DSB repair function. We found that in a *tel1Δ* background, *rad50-L1240F* and *mre11-E38K* phenocopied Mre11 complex null mutations with respect to CPT sensitivity at the highest doses used. *rad50-L1240F* showed partial synergism and *mre11-E38K* strong synergism with *tel1-kd* for survival on CPT. In contrast, *rad50-D67N/Y* and *rad50-R1259C* do not exhibit temperature sensitivity (S4 Fig) and their survival on CPT was only slightly affected by either the absence of the Tel1 protein or its kinase activity (Fig 4B). Therefore, these data indicate that *rad50-L1240F* and *mre11-E38K* depend on interaction with Tel1 protein for DSB repair, and their defect in Tel1 activation is attributable to an intrinsic molecular function. *rad50-D67Y/N* do not depend on Tel1 protein for DSB repair but the data similarly suggests a defect in an activity intrinsic to Rad50 required for Tel1/ATM activation.

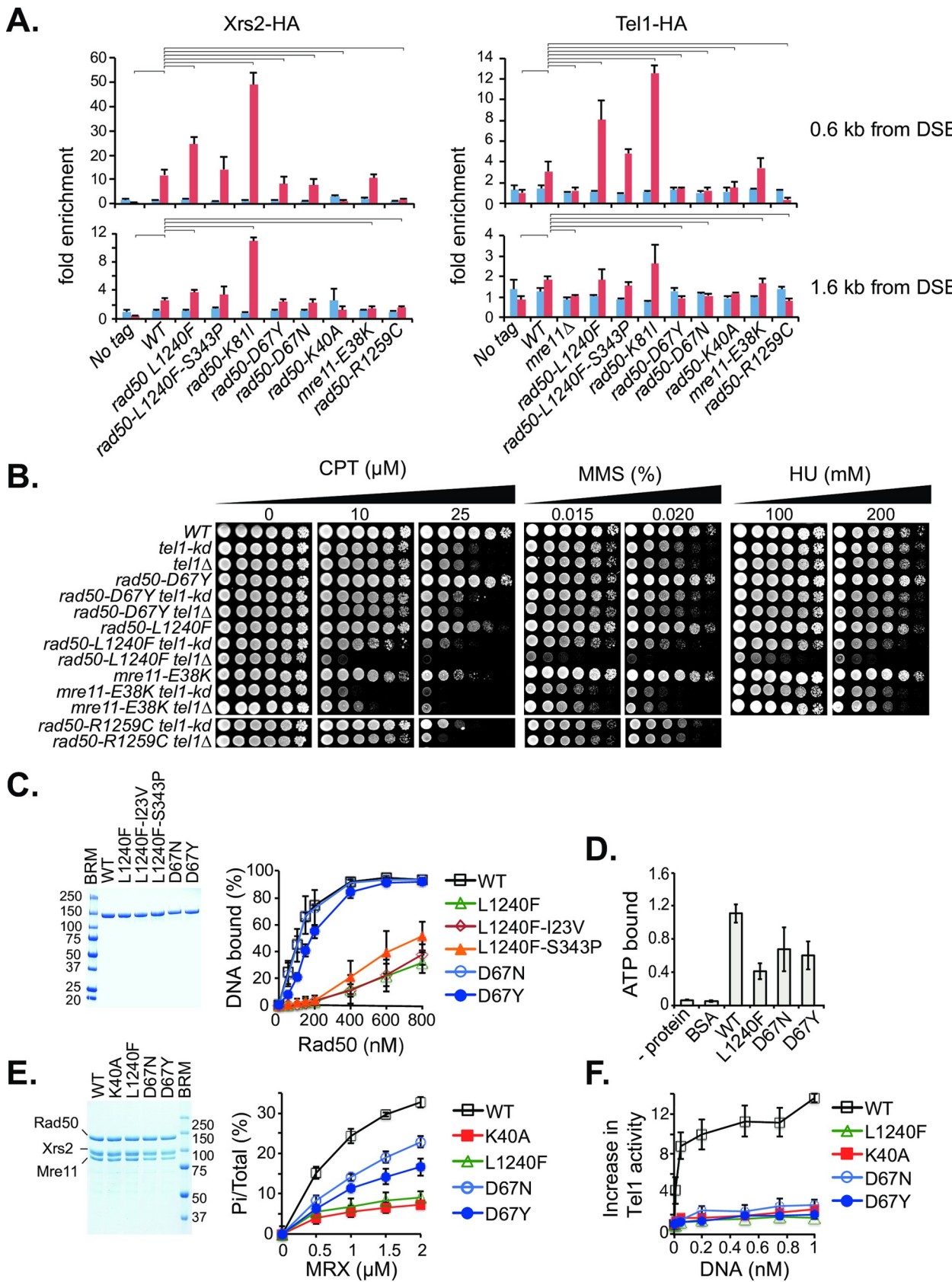

**Fig 4. Mre11 complex and Tel1 DSB-association.** (A) Xrs2 and Tel1 recruitment to HO-DSB after 0 hours (in blue) and 3 hours (in red) of galactose induction. Relative fold enrichment of Xrs2-HA and Tel1-HA at 0.6 kb (top) and 1.6 kb (bottom) from the HO-DSB were determined by CHIP and qPCR analysis. A no HA-tag control strain was included as negative control. Significant differences between WT and mutants are denoted above the graphs (one-tailed, unpaired Student's *t* test; p < 0.05). (B) CPT-, MMS- and HU- survival of modeled Mre11 alleles in absence of Tel1 kinase (*tel1-kd*) activity or in absence of Tel1 protein (*tel1Δ*). *rad50-R1259C tel1-kd* and *tel1Δ* were assessed on separate plates. (C) ATP-dependent dsDNA binding of purified Rad50 WT and mutant proteins by EMSA. 2 μg purified Rad50 proteins were separated on 4–20% SDS-PAGE gradient gel and stained by Coomassie Blue (left panel). Increasing concentrations of Rad50 (0–800 nM) were incubated in a binding buffer containing 150 mM NaCl with 5 nM of a $^{32}$P-labeled 83-mer dsDNA oligonucleotide in presence of ATP and MgCl2. DNA binding was also assessed at 50 mM, 250 and 300 mM NaCl (see S5 Fig). (D) Rad50 ATP binding. 2 μg of purified Rad50 WT and mutant proteins were incubated with 49.9 μM ATP (spiked with 0.1 μM α-$^{32}$P- ATP) and ATP binding was assessed by nitrocellulose filter binding assays. The graph denotes standard deviations from three independent experiments. No protein or BSA controls were included as negative controls. (E) Rad50 ATPase activity of WT and mutant Mre11 complexes. 2 μg of purified Mre11 complexes were separated on a 7.5% SDS-PAGE gel and stained by Coomassie Blue (left panel). The ATPase activity of WT and mutant Mre11 complexes was determined by incubation of increasing concentrations of MRX with γ$^{32}$P-ATP, followed by separation of the hydrolyzed γ-$^{32}$P from the nonhydrolyzed γ$^{32}$P-ATP by thin layer chromatography and visualized by Phosphorimager analysis (an example is given in S6A Fig). Scans were quantified to calculate the signal intensity of Pi versus total signal per lane. Error bars denote the standard deviation from 4 independent experiments. (F) *In vitro* Tel1-activation by MRX and DNA. Standard kinase reactions contained 5 nM Tel1, 30 nM MRX WT and mutant proteins, 200 nM Rad53-kd (kinase dead) and increasing concentrations of a 2 kb linearized plasmid DNA. Triplicate experiments were quantified. An example is given in S7 Fig.

## Biochemical analysis of SOF mutant gene products

Structural analyses of the Mre11 complex globular domain suggest that ATP binding and hydrolysis determine whether it adopts a closed or open form. The former is proposed to mediate DSB end tethering, NHEJ, and Tel1/ATM activation [5,37]. Transition to the open state depends on ATP hydrolysis by Rad50 in which the globular domain opens to make the Mre11 nuclease active sites accessible for DNA substrates [2,38,39].

*rad50-D67N/Y* and *rad50-L1240F* mutations alter residues in the Rad50 ATPase domain. Given that ATP binding is required for DNA binding [22], we purified the WT and mutant Rad50 proteins from yeast cells (Fig 4C, left panel) and the corresponding MRX-holo complexes from insect cells (Fig 4E, left panel). We carried out assessment of DNA and ATP binding, as well as ATP hydrolysis.

DNA binding was measured by electrophoretic mobility shift assay (EMSA) in the presence of increasing concentrations (0–800 nM) of WT and Rad50-D67N, Rad50-D67Y, Rad50-L1240F, Rad50-L1240F-I23V and Rad50-L1240F-S343P proteins. Rad50-D67N ($K_D$ = 0.20 ± 0.03 μM) exhibited similar affinity for a 83 bp dsDNA oligo as wild type Rad50 ($K_D$ = 0.18 ± 0.02 μM), while Rad50-D67Y showed partially reduced affinity ($K_D$ = 0.32 ± 0.07 μM) (Fig 4C). In contrast, DNA binding was significantly reduced in Rad50-L1240F and Rad50-L1240-F-I23V ($K_D$ >0.8 μM) whereas the S343P exhibited a subtle increase. Unlike WT and Rad50-D67Y/N, DNA binding by Rad50-L1240F was strongly decreased at higher salt concentrations (S5 Fig). The severe defect in DNA binding observed with Rad50-L1240F was unexpected given the ChIP data above (Fig 4A), and likely reflects the difference between Rad50 behavior alone (in DNA binding *in vitro*) and in complex *in vivo* where Tel1 also modulates the complex's behavior.

Rad50 ATP binding was measured using a filter binding assay. WT and mutant Rad50 proteins (2 μM) were incubated for 20 minutes at room temperature with a molar excess of ATP (0.1 μM α$^{32}$P-ATP + 49.9 μM unlabeled ATP). Following incubation, the reactions were spotted on nitrocellulose filters, washed, and bound radiolabelled ATP was quantified by liquid scintillation counting. Under these conditions, the molar ratio of bound ATP to Rad50 was 1.1, in agreement with structural determination of ATP:Rad50 stoichiometry [40]. ATP binding was reduced by approximately two fold in the Rad50 mutants; Rad50-D67N (0.6 ATP/ Rad50), Rad50-D67Y (0.7 ATP/Rad50) and Rad50-L1240F (0.4 ATP/Rad50) (Fig 4D).

To measure ATP hydrolysis, increasing concentrations (0–2 μM) of purified MRX WT and mutant complexes were incubated with γ$^{32}$P- ATP and MgCl$_2$ in presence of ssDNA to

stimulate Rad50 ATP hydrolysis [22], and hydrolysis was assessed by thin layer chromatography (Fig 4E and S6A Fig). We find that both MRX-D67Y and -D67N exhibited a 50–70% reduction in ATP hydrolysis relative to MRX-WT (Fig 4E), consistent with their respective decrements in ATP binding (Fig 4D). ATP-hydrolysis of MRX-L1240F and MRX-K40A was comparably low. MRX-L1240F-S343P showed modestly increased ATP hydrolysis compared to MRX-L1240F (S6B Fig) in accordance with its partial mitigation of the DNA binding defect of Rad50-L1240F (S5 Fig).

Recently we have established an assay to measure Mre11 complex- and DNA-dependent Tel1 activation *in vitro* [34]. To assess Tel1 activation by modeled mutant Mre11 complexes (Fig 4F), 30 nM of the purified Mre11 WT and mutant proteins were incubated with 5 nM Tel1 and 200 nM Rad53-kd (kinase dead) in presence of increasing concentrations of plasmid DNA (0–1 nM). Tel1-mediated Rad53-kd phosphorylation was measured at each DNA concentration and is shown as increase in Tel1 activity (Fig 4F and S7 Fig). Whereas the WT complex stimulated Tel1 kinase activity by >10-fold, Tel1 was stimulated only about 2-fold by Mre11 complex containing Rad50-D67N, Rad50-D67Y Rad50-L1240F, comparable to what is seen with Rad50-K40A.

Collectively these data suggest that ATM activation is selected against during the progression of malignancy. The common mechanistic underpinning of the observed separation of function phenotypes is related to ATP binding and hydrolysis. Other defects are unique; in the case of Rad50-L1240F, defects in DNA binding also are correlated with the defect in Tel1/ATM activation whereas in in Rad50-D67Y/N, the Tel1/ATM activation defect may be partially attributable to reduced Tel1 recruitment to DSBs.

## Structural insight of tumor modeled mutations derived by homology modeling and molecular dynamics simulation

Structural information regarding the eukaryotic Mre11 complex is available for the globular domain and the Rad50 hook domain [2]. We used globular domain information to carry out a combination of molecular modeling, bioinformatics and molecular dynamics simulations to gain insight regarding the molecular features of *rad50-D67N/Y*.

First, by homology modeling, a structure of the ATPγS-bound heterodimeric Mre11-Rad50 *S. cerevisiae* containing a short region of the coiled coil domain (amino acids 177–218 and 1102–1159) was built. Rad50-D67 in this structural model is positioned next to the adenine base of ATPγS (Fig 5A). Both L1240F and R1259C are positioned away from the ATP binding site (Fig 5A and S8 Fig) and were not considered further in this analysis.

Molecular dynamics simulation performed in MR complexes with Rad50-D67 (WT), -D67N and -D67Y show that both mutants alter the interaction between the nucleotide and its binding site (S1, S2 and S3 Movies). The D67 mutations resulted in increased interaction times and strength of the contacts between the adenosine base and N67/Y67, I65, and V63 (Fig 5B and 5C), while weakening contacts with K1193. The presence of a single carbonyl oxygen of N67 leads to a frozen conformation of its side chain mediated by the NH proton of K69 (Fig 5B). Therefore, the release of ADP upon hydrolysis may be impaired by the D67N and D67Y mutations (S2 and S3 Movies). We speculate that impaired release of ADP in N67/Y67 mutants may alter the dynamics of the transition from the closed to the open form. These findings lend some weight to the idea that the *transition* from closed to open, rather than the open state vs. closed state may be the key step in Tel1/ATM activation.

## Discussion

The Mre11 complex is required for all forms of DSB repair as well as for the initial detection of DSBs and the subsequent activation of the ATM axis of the DDR. Rapidly accumulating cancer

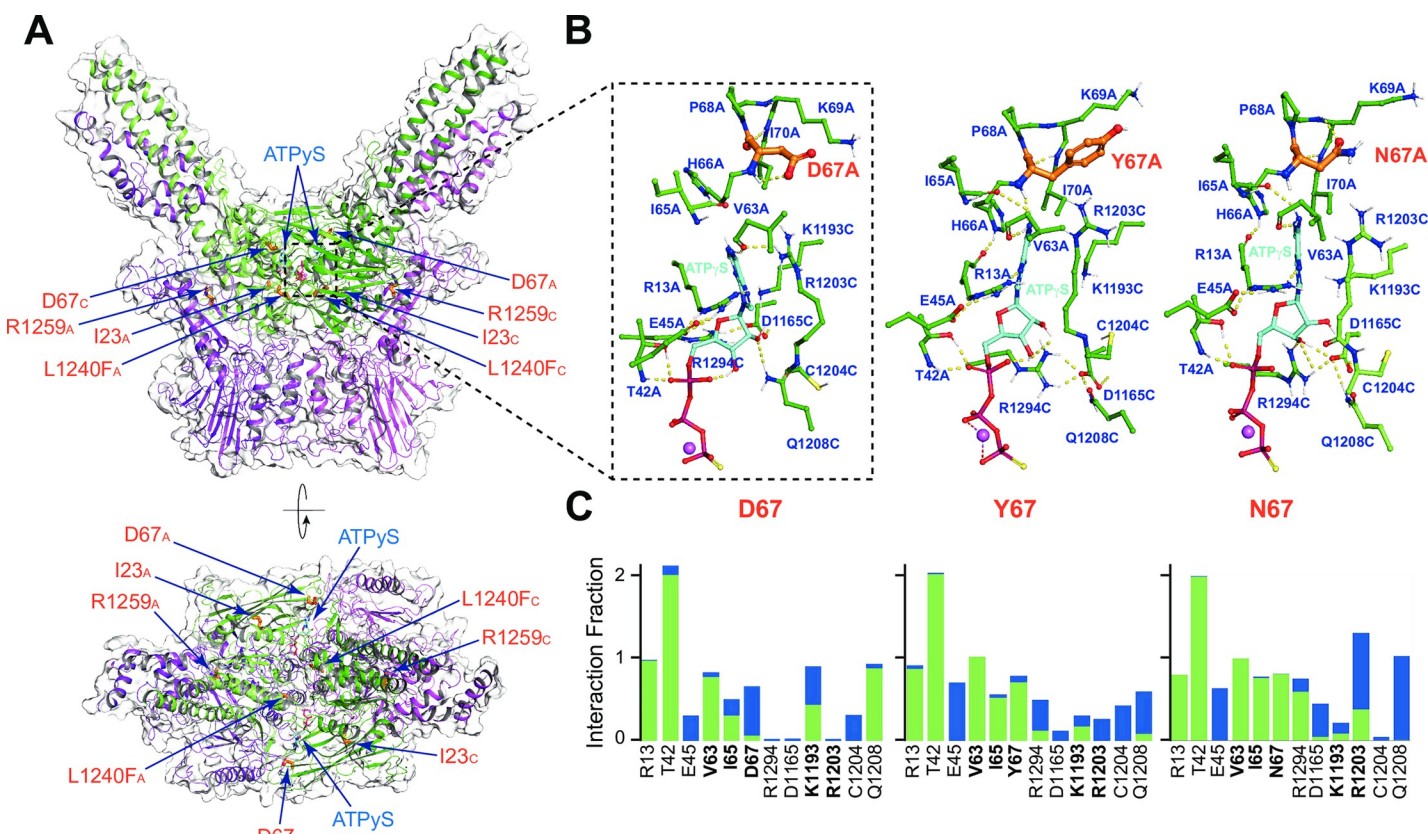

**Fig 5. Location of SOF mutations.** (A) Mutations are shown on a molecular model of the *S. cerevisiae* Mre11-Rad50 complex. Side and top views of Mre11-Rad50 complex shown as a transparent molecular surface with highlighted secondary structure elements; Mre11 colored magenta, Rad50 colored green. The wild type residues belonging to the chains A and C of Rad50 and affected by mutations in this study are shown as thick orange sticks, two ATPγS molecules are shown as ball-and-stick with carbons colored cyan. Coiled coils on the top view are optically partially truncated. PDB files of the images shown are available in the S1 File. (B) Zoomed views of the amino acid residue side chains interacting with ATPγS molecule at the end of MD simulation. The position of the zoomed view on panels A and B is indicated as a black dotted-lines box. (C) Residue-specific protein-ATPγS interactions over the entire course of MD simulation with the wild-type Rad50-D67 residue and Rad50-Y67/N67 mutant residues. Direct and water-mediated hydrogen bonds are colored green and blue, correspondingly. Amino acid residues that interact with the adenine base are highlighted in bold. Interaction fraction is equal to two for T42 because two hydrogen-bond donors of this residue are maintaining contacts with the non-bridging phosphate oxygen of ATPγS (panel B) throughout the course of the simulation. MD simulation movies are available in the S2 File.

genomic data has revealed that Mre11 complex components are mutated in approximately five percent of solid tumors [16]. This observation, along with data from humans and mouse model systems support the view that the Mre11 complex and the processes that it influences suppress tumorigenesis [1,14,41].

In this study, we used cancer genomic data to investigate the molecular mechanism(s) that underlie Mre11 complex-dependent tumor suppression using *S. cerevisiae* for genetic analysis as well as biochemical analyses of recurrent mutant gene products. The mutations modeled were selected on the basis of conservation of the affected residues, allele frequency in the tumor, and recurrence. Two themes emerged. First, the mutations modeled predominantly exhibited separation of function phenotypes characterized by severely impaired Tel1/ATM activation without substantial impairment of DSB repair. Second, the gene products of recurrent *RAD50* alleles found in seventeen distinct tumors exhibited ATP binding and/or hydrolysis defects. Taken together, these themes support the interpretation that selection against Mre11 complex-dependent ATM signaling occurs during tumor progression, and that Rad50-dependent ATP metabolism is crucial for ATM activation by the Mre11 complex.

It is important to note that we did not randomly sample all Mre11 complex mutations. Filtering the mutations for conserved residues implicitly biases the analysis to the Rad50 globular domain given its higher degree of conservation. Nevertheless, the recurrence of mutations that affect this domain provides a compelling argument for selective pressure.

Tel1/ATM activation requires DNA and the Mre11 complex, although the underlying molecular mechanism is unknown [42]. Each of the modeled *Rad50* SOF mutations severely impaired Tel1 activation *in vivo* and *in vitro* (Figs 3B, 3C and 4F). In principle, this phenotype could reflect loss of Mre11 complex DNA binding, loss of Mre11 complex-Tel1/ATM interaction, or impairment of an unknown mechanism.

Binding to naked DNA *in vitro* was not correlated with defective Tel1/ATM activation. Only Rad50-L1240F-containing complexes were impaired in this respect. Similarly, ChIP analysis revealed no clear correlation between Mre11 complex engagement at a DSB site and Tel1 kinase activation. These observations suggest that the underlying molecular bases of Tel1/ATM activation defects of Rad50-1240F and –D67Y/N are distinct.

A complex containing *rad50-K81I*, a *rad50S* protein that is hypermorphic for Tel1/ATM activation [43] exhibited increased DSB association and increased the abundance of Tel1 at the site as previously reported (Fig 4A) [33]. Conversely, Rad50-K40A-containing complexes, which do not bind ATP and phenocopy *rad50Δ* in most respects did not engage DSB sites or promote Tel1 association. However, Rad50-L1240F and -D67Y/N-containing complexes, which are both severely impaired in Tel1/ATM activation, had divergent effects on DSB engagement and Tel1 ChIP (Fig 4A and 4F).

Failure of the Mre11 complex to interact with Tel1 could also explain Tel1 activation defects. Tel1 ChIP signal was markedly reduced in *rad50-R1259C* and *rad50-D67Y/N*, while the ChIP signal of the corresponding mutant Mre11 complexes to DSB sites was evident (Fig 4A). The reduced ChIP signal may reflect impaired Mre11 complex-Tel1 interaction; however, in the context of *tel1-kd*, the sensitivity to high doses of CPT of *rad50-L1240F* and *rad50-R1259C* was mitigated in comparison to the *tel1Δ* background (Fig 4B). The mitigation would instead suggest that the inactive Tel1 protein binds to, and enhances the functionality of the Mre11 complex via physical interaction as proposed previously [10,36,44]. Consistent with this idea, the effect of *tel1-kd* on CPT sensitivity was most pronounced in the Rad50-L1240F mutant (Fig 4B), which binds DNA *in vitro* at least 10 fold less well than Rad50-D67N or -D67Y (Fig 4C). Thus, these data do not support the interpretation that the failure to activate Tel1 is attributable to impaired Mre11 complex-Tel1 interaction in the SOF mutants. The observation that CPT sensitivity in *mre11-E38K* strains was the same in *tel1-kd* and *tel1Δ* further argues that Tel1 protein primarily interacts with Rad50, as has been shown *in vitro* for ATM and human Rad50 [45].

ATP binding and hydrolysis by Rad50-L1240F-containing complexes is more severely impaired than–D67Y/N, while DSB engagement is enhanced relative to wild type and Rad50-D67Y/N. It therefore appears that ATP binding and hydrolysis are required for Tel1/ATM activation, and that this function is distinct from the recruitment of Tel1/ATM to DNA damage. We propose that ATP binding and hydrolysis underlie the mechanism of Tel1/ATM activation. And, that the defects shared by both classes of SOF mutants with respect to Tel1 activation reflect the ATP binding and hydrolysis defects observed in each class. Previous analysis of the human Mre11 complex suggested that the ATP bound "closed" form of the complex is responsible for ATM activation [37]. Our data are consistent with that view, although it is possible that the transition *per se* from the closed to the open form, which occurs upon ATP hydrolysis, underlies the mechanism of Tel1/ATM kinase activation rather than one form or the other. However, as ATM activation by the human Mre11 complex *in vitro* required ATP binding but not ATP hydrolysis [37], and ATP-binding induces multiple conformational

switches in both Rad50 and Mre11 prior to ATP-hydrolysis [40,46], it is tempting to speculate that a yet undefined conformational state between the closed and open complex mediates Tel1 activation. A recent study assessing the *rad50-L828F* archaeal mutation (orthologous to *L1240F*) suggests that the ATP bound closed state of Rad50 is disfavored [47]. It is also note-worthy that molecular dynamics simulation of ATP binding in the Rad50-D67Y or -D67N mutants suggests that the release of ADP following hydrolysis is impaired. It is thus conceiv-able that the transition between closed and open forms may be altered in that mutant, and may in turn account for the observed Tel1 activation defect.

Although the SOF mutants described were largely DSB repair proficient, a subtle effect on NHEJ was observed. NHEJ junctions in *rad50* SOF mutants predominantly exhibited deletions as opposed to the insertions typically observed in wild type cells (Fig 2C). The mechanistic basis for this difference is unclear, but we have previously noted a correlation between defects in Tel1 activation and deletional NHEJ. For example, in *rad50*<sup>sc+h</sup>, *rad50-48*, and *rad50ΔCC* mutants, which affect the Rad50 hook and coiled coil domains, deletions at NHEJ junctions are elevated. Like the tumor borne mutations described in this study, those *Rad50* alleles also exhibit SOF phenotypes in which Tel1 activation is impaired [6–8]. These observations reso-nate with analysis of NHEJ-mediated chromosomal translocations induced by HO cleavage. In Tel1 deficient cells, those junctions are characterized by the same *ΔACA* deletions observed at "unrepairable" DSBs in Mre11 complex deficient strains and in the SOF mutants described here and previously [48]. Although the various genetic contexts alluded to above have distinct phenotypic features, attenuation or abolition of Tel1 activation is common to each. These data thus support a role for Tel1 activity in influencing mechanistic features of NHEJ.

Why would ATM function be selected against during tumorigenesis? The role of the DDR in tumor suppression is multifaceted, and the prevailing view is that the DDR is an inducible barrier to oncogene driven carcinogenesis [49]. We have shown that the Mre11 complex-ATM axis of the DDR is crucial for this function. In mouse models, Mre11 complex hypomorphism enhances the ability of the *neuT* oncogene to promote malignancy in mammary epithelium [14], and promotes Notch-driven leukemogenesis in the hematopoietic compartment [50]. Collectively, those observations and those presented here support the view that selection against Mre11 complex-ATM signaling in the DDR potentiates oncogene driven carcinogene-sis to a greater extent than selection against the complex's DSB repair functions. Implicitly, the data further suggest that the loss of fitness that would be associated with reduced DNA repair capacity is also selected against during tumorigenesis. In contrast, the ATR-Chk1 axis of the DDR is required for viability of oncogene-expressing cells [51], excluding it from acting as a barrier. Finally, this study predicts that mutations impairing ATM activation or signaling may sensitize tumors to clastogenic therapies as well as those that inhibit the ATR-Chk1 axis of the DDR.

## Materials and methods

### Ethics statement

The 12–245 Data & Tissue Usage Committee at MSKCC has reviewed this study and has no comments or concerns.

### Yeast strains

All strains used in this study were in W303+ background and are listed in S3 Table. All *rad50* and *mre11* alleles in this study were integrated at their native chromosomal locus and verified by PCR genotyping and sequencing. Details of yeast strains and plasmid constructions are available upon request.

## Damage sensitivity assays

Five-fold serial cell dilutions (250,000 cells per spot to 80 cells per spot) were spotted on YPD plates without or with S-phase clastogens and incubated at 30˚C for 2 days unless stated otherwise.

## DSB repair by NHEJ

*rad50*::*HYG* and *mre11*::*kanMX* tumor modeled alleles were crossed in the W303-HO background (*leu2*::*GAL-HO-LEU2 HA-TEL1-URA hmlΔ hmrΔ Matα*) [36]. Spores were grown in yeast extract peptone medium containing 2.6% (v/v) glycerol, 2.6% (v/v) ethanol, 1% (v/v) succinate and 1% sucrose to exponential phase. Cells were counted and plated in triplicate on plates of identical composition additionally containing either 2% galactose or 2% glucose. Cell survival is expressed as the percentage of cells growing on galactose versus glucose containing plates after 4 days incubation.

## Rad53 phosphorylation

Rad53 phosphorylation was assessed as described previously [9]. Briefly, exponentially growing cells ($2-4\times10^7$ cells/ml) in *mec1Δ sml1Δ* or *mec1Δ sml1Δ sae2Δ* background were cultured in presence of 0.15% MMS for 90 min. MMS was inactivated upon addition of 5% sodium thiosulfate (final concentration) to the cultures. TCA-extracts were prepared and 10–20 μg protein extracts were run on a 7.5% SDS-PAGE, transferred to nitrocellulose membrane and FLAG-Rad53 was detected by western blot with FLAG M2 mAb (Sigma).

## Sporulation efficiency and spore viability

Diploids cells were grown overnight in YPD media, then diluted 20-fold in YPA media (yeast extract, 2% potassium acetate, 100 mg/l adenine). After 12 hours incubation in YPA media, cells were gently pelleted, washed with water and incubated for 2–3 days in sporulation media (1% potassium acetate, 100 mg/l adenine). The sporulation efficiency was calculated by the numbers of tetrads present among >400 of sporulated cells. Spore viability was determined by tetrad dissection of 20–40 tetrads. Two independent diploids for each genotype were assessed.

## Q-PCR based resection assay

Cells from each strain were grown overnight in 10 ml YPLG to a cell titer of $1\times10^7$ cells/ml. 15 μg/mL Nocodazole was added to arrest cells in G2/M. 2 hours post nocodazole addition to the media, 2.5 mL of the cells were pelleted as t = 0 and 2% galactose was added for HO-DSB induction. Other timepoints were taken at t = 1, 2 and 4 hours. Genomic DNA was purified using standard genomic preparation methods and DNA was re-suspended in 100 μl water. Genomic DNA was treated with 5.0 μg/ml RNase A for 45min at 37˚C. 2 μl of DNA was added to tubes containing CutSmart buffer with or without *Rsa*I restriction enzyme and incubated at 37˚C for 2 hours. Quantitative PCR was performed using the Applied Biosystem QuantStudio 6 Flex machine. PowerUp SYBR Green Master Mix was used to quantify resection at *MAT1* (0.15 kb from DSB) and *MAT2* (4.8 kb from DSB). Pre1 was used as a negative control. *Rsa*I cut DNA was normalized to uncut DNA as previously described to quantify the percentage of ssDNA per total amount of DNA [52]. Same primers were used for Q-PCR as previously published [29].

## Telomere Southern blot

Freshly dissected spores were grown for 30 generations of growth and genomic DNA isolated by standard phenol chloroform extraction using glass beads [53]. Genomic DNA was either *Pst*I-or *Xho*I-digested (as specified in Figure legends), run on 1.3% agarose gel and transferred on an Amersham Hybond-XL membrane (GE Healthcare) and detected by Southern blot with a $^{32}$P-labeled telomere-specific probe (5′-TGTGGTGTGTGGGTGTGGTGT-3′) as described [54].

## Rad50 overexpression and purification from yeast cells

The yeast galactose inducible expression plasmid, pR50.1 (2μ, GAL-PGK-RAD50, leu2-d), was a gift from Patrick Sung [21]. To facilitate protein purification, a C-terminal 1xFLAG tag (DYKDDDDK) was inserted by site directed mutagenesis. *LEU2* prototroph colonies were grown at 30˚C in 1 liter Do-Leu lactic acid (3%) glycerol (3%) media (pH 5.5) supplemented with 0.5% sucrose. Stationary phase cultures were induced with 2% galactose for 18 hours. Cell pellets were resuspended in 10 ml lysis buffer (25 mM Tris-Cl pH 7.6, 300 mM NaCl, 10% glycerol, 0.1% Igepal, 1 mM EDTA, 2 mM β-mercaptoethanol, 1 mM PMSF, 1 μg/ml Aprotonin, 10 mM Benzamidine, 1 μg/ml Chymostatin, 5 μg/ml Leupeptin, 0.7 μg/ml Pepstatin A). The cell suspension was snap-frozen into liquid nitrogen to form yeast "popcorn" and cryogenically ground using a Freezer/Mill (6 cycles of 3 min, 30 Hz). The powdered yeast cells were thawed, 20 ml lysis buffer (as above) was added and resuspended by pipetting to remove all clumps. The yeast extract was clarified by centrifugation (40 min 20,000 rpm, ss34 rotor) and incubated over night at 4˚C in presence of 0.8 ml anti FLAG M2 agarose beads (Sigma). The FLAG beads were washed 10x with 10 ml lysis buffer. Rad50-FLAG proteins were eluted with 5x 0.8 ml elution buffer (lysis buffer containing 100 μg/ml 3xFLAG peptide*).* 4 ml FLAG eluate was concentrated to 0.8 ml using Amicon Ultra-4 Centrifual Filters Ultracel-30K. Protein concentrations were determined by Lowry protein assay. 10 μl aliquots (≈10 μM concentration) were flash frozen in liquid nitrogen and stored at -80˚C.

## Mre11 complex expression and purification from Sf9 insect cells

1 liter of Sf9 insect cells grown in Spinner Cultures in BioWhitaker Insect-XPRESS media (Lonza) supplemented with 5% FBS (Sigma) and 0.5% Anti-Anti (100X, Gibco) were transferred to a 2 liter Erlenmayer flask and co-infected with an optimized ratio of freshly amplified baculoviruses expressing Mre11-his6, Xrs2-FLAG and Rad50 WT (gift from Petr Cejka) or mutant proteins and incubated at 27˚C for 68 hours on a Innova 44 shaker at 120 rpm. Infected cells were harvested by centrifugation at 1000 x g, washed once with 1xPBS and resuspended by pipetting in 25 ml ice cold lysis buffer without salt (50 mM Tris-Cl pH 7.6, 10% glycerol, 2 mM β-mercaptoethanol, 0.05% Igepal, 1mM PMSF, 1 μg/ml Aprotonin, 10 mM Benzamidine, 1 μg/ml Chymostatin, 5 μg/ml Leupeptin, 0.7 μg/ml Pepstatin A). After 20 min on ice, 1/9 volume of 5 M NaCl was added (final concentration 0.5 M NaCl), along with 2 μl (≈ 500 U) Benzonase Nuclease (Sigma) and lysates were incubated for 20 min at 4˚C on a SCILOGEX MX-T6-S Analog Tube Roller at 35 rpm. The lysates were sonicated on ice using a microtip for 2x 30 sec at 30% amplitude, continuous sonication and centrifuged for 45 min at 20,000 rpm (ss34 rotor). The clarified cell lysate (about 40 ml) was incubated with 0.8 ml anti FLAG M2 agarose beads (Sigma) at 4˚C overnight on a tube roller at 10 rpm. FLAG-beads were collected by centrifugation (800 x g, Eppendorf Centrifuge 5810 R), and washed 10 x with 10 ml lysis buffer containing 500 mM NaCl. FLAG-elution was carried out in lysis buffer containing 250 mM NaCl and 100 μg/ml 3xFLAG-peptide, with four stepwise FLAG-peptide elutions, each for 1 hour at 4˚C on rotary. The pooled 4 ml FLAG eluate was concentrated to 0.5

ml using Amicon Ultra-4 Centrifual Filters Ultracel-30K. Protein concentrations were measured by Lowry protein assay using BSA as a standard and were typically > 4 mg/ml (>6 μM). 10 μl aliquots were flash frozen in liquid nitrogen and stored at -80˚C.

## Electrophoretic Mobility Shift Assay (EMSA)

Increasing amounts of Rad50 proteins (0–800 nM) were incubated with 5 nM of $^{32}$P-radiolabelled 83-bp double stranded DNA substrate [6] in 10 μl of DNA binding buffer (25 mM Tris-Cl 7.6, 150 mM NaCl, 7.5% glycerol, 1 mM β-mercaptoethanol, 0.125% Igepal, 5 mM MgCl2 as indicated either with 5 mM ATP or without ATP) for 20 min at RT and then 20 min on ice. Samples were loaded on 5% native 0.25x TBE PAA gel supplemented with 5 mM MgCl2, and run at 4˚C at 100V for 2 hours in running buffer (0.25x TBE+ 5 mM MgCl2). Gel were fixed for 15 min (25% Ethanol, 3% Glycerol) and dried on Grade 1 Chr cellulose chromatography paper (Whatman) and visualized by phosphorimager analysis. Gels were quantified using ImageGauge (GE) software and the percentage DNA bound determined by the ratio of signal intensity of the Rad50 bound dsDNA substrate versus total radioactivity per lane.

## ATPase assay

Purifed WT and Rad50-mutant Mre11 complexes (0–2 μM) were incubated in ATPase buffer (25 mM Tris-Cl pH 8.0, 100 mM NaCl, 1 mM DTT, 5 mM MgCl2, 0.1 mg/ml BSA, 0.1 μM γ-$^{32}$P-ATP, 0.5 mM cold ATP, 25 μM ssDNA (a 50 bp long oligonucleotide) in a reaction volume of 10 μl. After 90 min at 30˚C, the reaction was stopped by addition of 10 μl stop buffer (1% SDS, 10 mM EDTA) and 0.5 μl was loaded on a thin layer chromatography (TLC) plate (TLC PEI Cellulose F, EMD Millipore). The air-dried TLC plate was developed in a mobile phase of 0.5 M lithium chloride and 0.5 M formic acid and exposed to a phosphor-imager screen. The percentage of ATP hydrolysis was calculated as ratio of released Pi/total radioactivity per lane following quantification with ImageGauge software (GE).

## ATP binding assay

Rad50 ATP binding was determined by standard nitrocellulose filter binding assays. Briefly, 2 μM of purified Rad50 proteins were incubated with 50 μM ATP (containing 0.1 μM of radioactive α-$^{32}$P-ATP) in presence of binding buffer (25 mM Tris-Cl 8.0, 100 mM NaCl, 0.1% Igepal, 2 mM MgCl2, 1 mM β-mercaptoethanol) in a volume of 20 μl at RT for 20 min and then for 30 min on ice. The binding reaction was then loaded on a binding buffer presoaked nitrocellulose filter (MF-Millipore Membrane Filter, 0.45 μm pore size), and washed with gentle suction with 2 ml binding buffer. α-$^{32}$P-ATP retained on Nitrocellulose filters was measured by standard liquid scintillation counting. Serial dilutions of the input 50 μM ATP (containing α-$^{32}$P-ATP) were prepared and measured to determine the amount of ATP bound by Rad50.

## *In vitro* Tel1 kinase assay

Tel1 kinase assay was carried as previously described [34]. Briefly, 5 nM Tel1 was incubated with 30 nM MRX, 200 nM Rad53-kd and 0–1 nM of linearized plasmid DNA (2 kb) for 15 min at 30˚C in buffers specified previously.

## Chromatin Immunoprecipitation

Cells were cultured overnight in YPLG (1% yeast extract, 2% bactopeptone, 2% lactic acid, 3% glycerol and 0.05% glucose) at 25˚C. Cells were then diluted to equal levels (5 x 10$^6$ cells/ml) and were cultured for one doubling (3–4 hours) at 30˚C to 1 x 10$^7$ cells/ml. 2% galactose (final

concentration) was added to the YPLG and incubated for 3 hours. Cells were harvested and crosslinked at various time points (t = 0 hours and t = 3hours after galactose treatment) using 3.7% formaldehyde solution. Following crosslinking, the cells were washed with ice cold 1xPBS and the pellet stored at -80˚C. The pellet was re-suspended in lysis buffer (50 mM HEPES pH 7.5, 1 mM EDTA, 80 mM NaCl, 1% Triton, 1 mM PMSF and protease inhibitor cocktail) and cells were lysed using Zirconia beads and a bead beater. Chromatin fractionation was performed to enhance the chromatin bound nuclear fraction by spinning the cell lysate at 13,200 rpm for 15 minutes and discarding the supernatant. The pellet was resuspended in lysis buffer and sonicated to yield DNA fragments (~500 bp in length). The sonicated lysate was then incubated with Dyna beads (Sheep anti-Mouse IgG from Invitrogen) with anti-HA Anti-body (Santa Cruz) or unconjugated beads (control) for 2 hours at 4˚C. The beads were washed using wash buffer (100 mM Tris-Cl pH 8, 250 mM LiCl, 0.5% NP-40, 1 mM EDTA, 1 mM PMSF and protease inhibitor cocktail) and protein-DNA complexes were eluted by reverse crosslinking using 1% SDS in TE buffer, followed by proteinase K treatment and DNA isolation via phenol-chloroform-isoamyl-alcohol extraction. Quantitative PCR was performed using the Applied Biosystem QuantStudio 6 Flex machine. PerfeCTa qPCR SuperMix, ROX was used to visualize enrichment at HO2 (0.6 kb from DSB) and HO1 (1.6 kb from DSB) and *SMC2* was used as an internal control. The Ct values from the qPCR were used to estimate the amplification of the region at 0.6 kb (HO2) and 1.6 kb (HO1) from the DSB site. The enrich-ment values were obtained by normalizing the values of HO1 and HO2 by values obtained from *SMC2* gene locus. Following qPCR primers were used: HO2-F (TTGCCCACTT CTAA GCTGATTTC), HO2-R (GTACTT TTCTACATTGGGAAGCAA TAAA), HO2 Probe (FAM-ATGATGTCTGGGTTTTGT TTGGGATGCA-TAMRA); HO1-F (GTTCTCATGC TGTCGAGGATTTT), HO1-R (AGACGTCCTTCTACAACAA TTCATAAGT), HO1 Probe (FAM-TTTGGGACGAT ATTGTCATTATAGGGCAGTG TG-TAMRA); SMC2-F (AATTG GATTTGGC TAAGCGTAATC), SMC2-R (CTCCAAT GTCCCTCAAAATTTCTT), SMC2 Probe (FAM-CGACGCGAATCCATCTTCCCAA ATAATT-TAMRA).

## Homology modeling and molecular dynamics simulation

*S. cerevisiae* MR complex has been generated by homology modeling of the constituent Mre11 (M) and Rad50 (R) molecules and their subsequent assembly in M2/R2 heterotetramer. Sec-ondary structure predictions from amino acid sequences, access numbers CAA65494.1 and BAA02017.1 [55], were computed by PSIPRED [56,57].

## Monomeric Mre11

*S. cerevisiae* Mre11 was built with Schrödinger Prime homology package [58] using *S. pombe* crystal structure as a template (RCSB entry 4FBQ.pdb: resolution 2.2 Å, sequence homology and identity 68% and 51%, respectively), and guided by secondary structure predictions. The model has been edited to enhance its similarity with hsMre11 (RCSB entry 3TI1.pdb)

## Generation of scMre11 dimer

Two metal ions introduced into the homology model from hsMre11 structure were converted to dummy ligands. Similar dummy ligands were generated on each of two scMre11 monomers of the MR crystal structure from archaea *M. jannaschii* (RCSB entry 5DNY.pdb) [38]. These artificial ligand binding sites were used for two subsequent applications of Align Binding Sites tool [58], thus generating a dimeric assembly of scMre11. The conformations of the contacting loops at the interface of two scMre11 molecules have been edited by copying corresponding conformations from the crystal structure of *C. thermophilum* (RCSB entry 4YKE.pdb) [59].

The conformations of the loops protruding towards presumed scRad50 binding interface have been copied from the corresponding parts of the archean M2/R2 assembly (RCSB entry 5DNY.pdb).

## Monomeric Rad50 of S. cerevisiae

The initial models for monomeric Rad50 were generated by SWISS-MODEL [60], using 5DA9.pdb structure from *C. thermophilum* as a template [59].

## Generation of M2/R2 heterodimers

Initial positioning of scRad50 over scMre11 was obtained by pairwise aligning lobe I c-alpha atoms [39] of scRad50 and mjRad50 by MR heterodimer structure of *M. jannaschii* (RCSB entry 5DNY.pdb). After that, ATP analog ligands were introduced into scRad50 monomers by aligning its protein c-alpha atoms with these of ctRad50. The new positions of Rad50 were subsequently improved by the application of Align Binding Sites tool, targeting one pair of ligands at a time (one from scRad50, one from ctRad50). Both molecules of scRad50 were next superimposed as a dimer onto the ctRad50 dimer by all c-alpha atoms. One more iteration of monomer-monomer re-adjustment and re-introduction of ATP analog ligands by c-alpha atoms have improved the overall architecture of the complex and the positioning of ATP analogs in it. A few close contacts were removed by selecting appropriate rotamers of amino acid residues using Maestro graphical user interface of Schrodinger [58]. The resulting complex had no bad contacts at all; it was next energy-minimized and additionally cleaned up by 10 ns molecular dynamics.

## Symmetry enhancements

Two Mre11+Rad50 halves of the scMR complex cleaned up with energy minimization and molecular dynamics were merged into a single molecule each. A copy of one such heterodimeric half was superimposed onto another one by all its c-alpha atoms. Rad50 constituent molecules of these halves were next superimposed individually, without Mre11, and the new positions became parts of the updated complex. The ATP analogs were re-introduced in this updated complex after one more round of superimposing of a dimeric ctRad50 on the total scMR dimer, with the subsequent adjustments of monomeric scRad50 molecules onto each monomeric ctRad50.

## Introduction of RBD helices

RBD helices were homology modeled on ctMre11 (5DA9.pdb). Their position on coiled coils of scMR model was obtained by the superposition of coiled coils of ctMR and scMR.

The missing loop between K410 and L444 that connects RBD helices with Mre11 has been generated in two steps. Initially, a fragment from the RCSB entry 1DY0.pdb identified using NGL-SuperLooper [61] was introduced (A chain, residues 192–223). The standard alpha-helical six-residue segment predicted by PSIPRED for D430-K436 of scRBD domain. A shorter remaining gap between D430 and K410 was then filled with PrimeX module of Schrödinger suite.

The resulting protein structure was subjected to Protein Wizard of Schrödinger suite, energy minimizations, and molecular dynamics.

### Introduction of mutants

Rad50-D67N, -D67, -I23V, and -R1259C mutants were obtained by Mutate Residue tool implemented in Maestro graphical user interface of Schrödinger suite. Accommodation of Y67 in the structure required conformation adjustments of the neighboring residues M102, L1198, and R1203. Two conformers of Y67 side chain have been reviewed: one directed outwards and one inwards, with the aromatic ring in the inward conformation being parallel to the adenine base of ATP ligand. For the inward mutation, the rotamers of T107, M1191, L1198, and R1203 were screened to select the one with the minimal clashes. All mutant complexes were minimized with conjugate gradient minimization for $2x10^4$ steps.

Rad50-L1240F mutants have their side chains positioned close to each other at the dimer interface and could potentially interfere with the Rad50 dimer assembly. In the presence of ATP molecules, the Prime module of Schrödinger suite could not fill in an artificially generated gap of three residues around residue L1240. The studied structure of the Rad50-L1240F mutants was thus obtained by fitting the bulkier phenylalanine side chains at the interface already preformed, and with two ATP analog molecules already bound.

### Molecular dynamics

MD simulations were run for 10 to 100 ns with OPLS3e forcefield implemented as Desmond module [62] in Schrödinger suite, versions 2018–3 through 2019–1. Protein complexes surrounded by a 10Å buffer in an orthorhombic box solvated in SPC water and with charges neutralized were prepared with System Builder of Maestro GUI. The relaxation protocol before production run included two steps of minimization followed by four MD runs. The first two runs 12 ps each were performed at 10K and with short 1 fs timesteps (as an NVT ensemble first and as an NPT ensemble second), with all solute heavy atoms fixed. Two subsequent runs as NPT ensembles were at 300 K and with normal 2 fs timesteps: first for 12 ps with solute heavy atoms fixed, followed by a 24 ps run with no restraints imposed.

A MD production run of 10 to 100 ns was carried out at a constant temperature and pressure with a 2 fs timestep throughout the simulation. The thermostat Nose-Hoover chain method was applied with relaxation time of 1 ps [63]. Barostat parameters were set according to Martyna et al. [64] with a relaxation time of 2.0 ps with isotropic coupling. A 9 Å cutoff was applied to Lennard–Jones interactions, and the nonbonded list was updated every 1.2 ps. The production snapshots of the coordinates were written out every 1.2 ps.

The data were analyzed by the Simulation Event Analysis and Simulation Interaction Diagram modules implemented in Schrödinger suite of programs.

### Movies

Movies were prepared with Maestro by exporting frames from an entire 10 ns MD trajectory, 0.12 sec per frame.

### Supporting information

**S1 Fig. DSB repair and resection is largely intact in SOF mutants.** Figure related to Fig 2. (A) Mating type switching assay to measure HO-DSB by homologous recombination. HO-DSB formation was induced for one hour in *MAT***a** cells and HO-DSB repair was monitored by southern blot over 4 hours using *Sty*I-digested genomic DNA and a *MAT* locus specific probe (chromosome III coordinates 201176 to 201580) as previously described [65]. The HO-endonuclease produces a 0.7 kb fragment (HO-cut, indicated by arrow) from the 0.9 kb *MAT***a** *Sty*I fragment (indicated by an arrow) and a 1.8 kb *Sty*I-fragment is produced upon

homologous recombination repair of the HO-DSB with the *MATα* donor template (indicated by an arrow). Supporting Methods are given in S1 Text. (B). Clastogen survival of *rad50-L1240F*, *rad50-D67Y* and *mre11-E38K* in *sgs1Δ* and/or *exo1Δ* background. For comparison, *tel1Δ* and *mre11* nuclease dead alleles *(mre11-H125N* or *mre11-3)* were included.
(TIF)

**S2 Fig. Rad53 phosphorylation upon MMS-treatment assessed by Phos-tag gel electrophoresis of Mre11 complex modeled mutants in *mec1Δ* and *mec1Δ sae2Δ* background.**
Figure related to Fig 3B. Exponentially growing cells were treated for 90 min with 0.15% MMS and 10 μg TCA-extracts were run on a 8% PAA gel (14x16 cm) containing 20 μM Phos-tag (Fujifilm Wako Pure Chemical Corporation, AAL-107) and 40 μM MnCl2 for 16 hours at 50V at RT. The gel was soaked for 20 min in 10 mM EDTA, transferred to PVDF membrane and probed sequential with FLAG M2 antibody and anti-mouse HRP. A short and long exposure of the same membrane is shown.
(TIF)

**S3 Fig. Phenotypic characterization of *rad50-L1240F* intragenic suppressors.** Please see also S2 Text for a more detailed description. (A) MMS- and CPT-survival of *rad50-L1240F* suppressor mutants S343P, I23V, A1079N and S1247N in *mec1Δ* and *mec1Δ sae2Δ* background. (B) Identified *rad50-L1240F* intragenic suppressors denoted on Rad50 primary structure. (C) Tel1-dependent Rad53 phosphorylation in *mec1Δ* and *mec1Δ sae2Δ* cells after 90 min treatment with 0.1% MMS (+) was assessed by western blotting with anti-Flag-Rad53. Migration levels of the non-phosphorylated form (Rad53) and the phosphorylated form (P-Rad53) are indicated. Accordingly to survival in *mec1Δ* and *mec1Δ sae2Δ* background, the suppressor subtly mitigated the Tel1-dependent Rad53-phoshorylation defect of *rad50-L1240F*. (D). Telomere southern blots. Effect of *rad50-L1240F* intragenic suppressors on telomere lengths. Telomere lengths were assessed of freshly dissected spores after 30 generation of growth with either *Pst*I digested (blot on right) or *Xho*I-digested (blot on left) genomic. Size makers are only given for one of the blots. (E) MMS- and CPT-survival of *rad50-L1240F* intragenic suppressors in Mec1-proficient background. (F). *rad50-L1240F* partial meiotic phenotypes is suppressed to *WT* levels in *rad50-L1240F-S343P* and *rad50-L1240F-I23V* diploids. Sporulation efficiency (left axis) and spore viability (right axis) are plotted. (G) Mre11 complex integrity of *rad50-L1240F* without and with intragenic suppressors assessed by Rad50 and Mre11 immunoprecipitation and western blotting. (H) Q-PCR based resection assay. The S343P suppressor alleviates *rad50-L1240F* reduced DSB resection. Error bars denote standard deviation from three experiments. Other suppressors were not assessed.
(TIF)

**S4 Fig. *rad50-L1240F* and *mre11-E38K* are temperature sensitive in CPT survival.**
Figure related to Fig 2A. Indicated strains were incubated at 30˚C or 37˚C for 2 days or at 23˚C for 4 days.
(TIF)

**S5 Fig. Salt-dependence of Rad50 ATP-dependent DNA-binding.** Figure related to Fig 4C. Rad50 dsDNA binding was assessed was assessed at 50 mM, 150 mM, 250 mM and 300 mM NaCl. Increasing concentrations of Rad50 (0–800 nM) were incubated in a binding buffer containing the indicated concentrations of NaCl with 5 nM of a $^{32}$P-labeled 83-mer dsDNA oligonucleotide in presence of ATP and MgCl2 or absence of ATP (assessed only for 800 nM Rad50). The migration levels of the unbound (u) and Rad50 bound (b) DNA substrate is denoted. A quantification of the shown EMSA gels is given (on bottom).
(TIF)

**S6 Fig. Rad50 ATPase activity of MRX-WT and–mutant complexes assessed by thin layer chromatography.** Figure related to Fig 4E. (A) ATPase activity of modeled mutants. Increasing concentrations of MRX complexes (0–2 μM) were incubated with $\gamma^{32}$P-ATP- in presence of ssDNA and samples were run on a TLC plate. The migration levels of the hydrolyzed free phosphate ($\gamma^{32}$P) and the non-hydrolyzed ATP (ATP-$\gamma^{32}$P) are indicated. The signal intensity of $\gamma^{32}$P and total signal per lane was quantified and the percent ATP hydrolysis (Pi/total) is given (bottom of TLC plate). Four independent experiments were quantified and are illustrated in graph shown in Fig 4E. (B) ATPase activity of Rad50-L1240F without and with I23V and S343P suppressors. 2 μg of purified Mre11 complexes were loaded on SDS-PAGE stained with Coomassie Blue. An example of an ATPase assay is shown. Standard deviations represent three experiments.
(TIF)

**S7 Fig. Mre11 complex- and DNA- dependent activation of Tel1 kinase.** Figure related to Fig 4F. Standard kinase reactions contained 200 nM Rad53-kd and 50 μM [$\gamma^{32}$P]-ATP in kinase buffer with or without 30 nM Mre11 complex and the indicated concentration of 2 kb linear DNA. Kinase reactions were initiated with 5 nM Tel1. Reactions were stopped after 15 min at 30 ˚C and analyzed by 7% SDS-PAGE, followed by phosphorimaging.
(TIF)

**S8 Fig. An indirect impact on the dynamics of *S. cerevisiae* Mre11-Rad50 homology model exerted by the mutants R1259C, L1240F, and L1240F+I23V.** (A) Rad50-R1259-mediated interactions between the RBD domain of Mre11 (magenta) and Rad50 (green) molecules. (B) Impact of Rad50-R1259C mutant on interactions between the RBD domain of Mre11 (magenta) and Rad50 (green) molecules. (C) Rad50-L1240F mutation is localized far from the residues directly interacting with the adenine base, but closer to the triphosphate binding site. (D) The overall Mre11-Rad50 dynamics indirectly mediates the rescue effect of I23V on Rad50-L1240F mutant function. (E) Residue-specific protein-ATP interactions over the entire course of MD simulation in the vicinity of the Rad50-D67 residue. Amino acid residues that interact with the adenine base are highlighted in bold.
(TIF)

**S1 Table. *Rad50* and *Mre11* tumor alleles modeled in yeast.** The table list all alleles modeled in yeast. Separation of function (SOF) alleles (described in main text) are highlighted on dark blue (strong SOF alleles) or light blue (partial SOF alleles) background. Alleles deficient in DSB-repair (*rad50Δ* and *mre11Δ* alike, severe MMS-sensitivity in *MEC1* background) are given on a red background. Alleles with only mild or no MMS-sensitivity are highlighted on a green background. Allele frequencies, number of mutations and tumor types are given. The alleles tested in this study are listed on the left side of the table. Some residues were also mutated in tumors to other amino acid residues (alleles given on right side of the table), but were not assessed in this study.
(TIF)

**S2 Table. Extended table of modeled *Rad50* and *Mre11* tumor alleles.** Related to Fig 1A. This is an extended version of above the table shown in Fig 1A, listing all allele frequencies, number of mutations present in tumor samples and sequencing database source.
(TIF)

**S3 Table. *S. cerevisiae* strains used in this study.**
(TIF)

**S1 File. Related to Fig 5A.** PDB files show the molecular model of the *S. cerevisiae* Mre11-Rad50 WT (Rad50-D67) and mutant (Rad50-D67N and Rad50-D67Y) complex. (ZIP)

**S2 File. Raw data of all graphs and summary statistics.** (ZIP)

**S1 Movie. Related to Fig 5B.** Molecular dynamics simulations performed on the wild type Mre11-Rad50 (MR) complex. For best performance use the VLC media player to play the movies. (MPG)

**S2 Movie. Related to Fig 5B.** Molecular dynamics simulations performed on the Rad50-D67N mutant MR complex. For best performance use the VLC media player to play the movies. (MPG)

**S3 Movie. Related to Fig 5B.** Molecular dynamics simulations performed on the Rad50-D67Y mutant MR complex. For best performance use the VLC media player to play the movies. (MPG)

**S1 Text. Two supporting methods are given.** (DOCX)

**S2 Text. Intragenic Suppressors of *rad50-L1240F*.** Supporting text for S3 Fig. (DOCX)

## Acknowledgments

We are grateful to Jim Haber (Brandeis University), Patrick Sung (Yale University), Lorraine Symington (Columbia University), Scott Keeney (MSKCC), Chris Lima (MSKCC), Petr Cejka (IRB, Switzerland) and the Zhao lab (MSKCC) for yeast strains, plasmids, reagents and technical support. We thank Elizabeth Swisher (UW Medicine), Maria Jasin (MSKCC) and the Ovarian Stand Up To Cancer (SU2C) Dream Team to communicate the *MRE11-E42K* ovarian cancer allele. We thank current members of the Petrini laboratory and Tom Kelly for helpful discussion throughout the course of this work. We thank Laura Feeney and Tom Kelly for critical reading of the manuscript.

## Author Contributions

**Conceptualization:** John H. J. Petrini.

**Data curation:** Marcel Hohl, Aditya Mojumdar, Sarem Hailemariam, Vitaly Kuryavyi, Fiorella Ghisays, Kyle Sorenson.

**Formal analysis:** Marcel Hohl, Aditya Mojumdar, Sarem Hailemariam, Vitaly Kuryavyi, Fiorella Ghisays, Kyle Sorenson.

**Funding acquisition:** Peter M. Burgers, Jennifer A. Cobb, John H. J. Petrini.

**Investigation:** Marcel Hohl, Aditya Mojumdar, Sarem Hailemariam, Vitaly Kuryavyi, Fiorella Ghisays, Kyle Sorenson.

**Methodology:** Marcel Hohl, Aditya Mojumdar, Sarem Hailemariam, Vitaly Kuryavyi, Matthew Chang, Barry S. Taylor, Peter M. Burgers, Jennifer A. Cobb, John H. J. Petrini.

**Resources:** Matthew Chang, Barry S. Taylor, Dinshaw J. Patel, Peter M. Burgers, Jennifer A. Cobb.

**Supervision:** Dinshaw J. Patel, Peter M. Burgers, Jennifer A. Cobb, John H. J. Petrini.

**Visualization:** Marcel Hohl, Aditya Mojumdar, Sarem Hailemariam, Vitaly Kuryavyi, Fiorella Ghisays.

**Writing – original draft:** Marcel Hohl, John H. J. Petrini.

**Writing – review & editing:** Marcel Hohl, John H. J. Petrini.

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
