## [Decision Letter · Decision Letter 0]

18 Oct 2019

Dear Dr Petrini,

Thank you very much for submitting your Research Article entitled 'Modeling Cancer Genomic Data in Yeast Reveals Selection Against ATM Function During Tumorigenesis' to PLOS Genetics. Your manuscript was fully evaluated at the editorial level and by independent peer reviewers. The reviewers appreciated the attention to an important problem, but raised some substantial concerns about the current manuscript. Based on the reviews, we will not be able to accept this version of the manuscript, but we would be willing to review again a much-revised version. We cannot, of course, promise publication at that time.

Should you decide to revise the manuscript for further consideration here, your revisions should address the specific points made by each reviewer.  The point raised by Reviewer 1 regarding whether several of the mutations characterized, e.g., rad50-L1240F and mre11-E38K, are clean separation-of-function mutations is a valid one in light of the synergism for DNA damage sensitivity with tel1-kd and the weak telomere length defect.  The decreased spore viability and increase in MRX binding at DSBs seen for rad50-L1240F could be due to a partial defect in Mre11 nuclease activation.  The suggestion that the signaling function of the Mre11 complex is more important for tumor suppression than its role in DNA repair seems over-stated when some of the mutations are hypomorphic for several Mre11 complex functions. We will also require a detailed list of your responses to the review comments and a description of the changes you have made in the manuscript.

If you decide to revise the manuscript for further consideration at PLOS Genetics, please aim to resubmit within the next 60 days, unless it will take extra time to address the concerns of the reviewers, in which case we would appreciate an expected resubmission date by email to plosgenetics@plos.org.

[LINK]

We are sorry that we cannot be more positive about your manuscript at this stage. Please do not hesitate to contact us if you have any concerns or questions.

Yours sincerely,

Lorraine S. Symington

Associate Editor

PLOS Genetics

Gregory P. Copenhaver

Editor-in-Chief

PLOS Genetics

Reviewer's Responses to Questions

**Comments to the Authors:**

Reviewer #1: The Mre11 (Mre11-Rad50-Xrs2) complex is required for DSB repair and activation of Tel1/ATM kinase. This study uses the yeast S. cerevisiae to model a number of tumor-derived Mre11 and Rad50 mutations to determine whether these mutations preferentially impact one or the other activity. The central finding from this study is that a “significant fraction of tumor-borne RAD50 and MRE11 mutations have separation of function (SOF) phenotypes wherein Tel1/ATM activation was defective while DNA repair functions were mildly or not affected.” These observation lead to the conclusion “that the signaling functions of the Mre11 complex are important for tumor suppression to a greater degree than its role in DNA repair.” If true, these conclusions are an important scientific advance.

There are a lot of interesting data in this manuscript. Bravo for the team effort and careful execution of these experiments. However, I am far from convinced by these data. The key problem is that none of the alleles are unambiguous SOF mutations. As but one example, the rad50-L1240F allele, classified as a strong SOF in this and a previous study, clearly confers intermediate MMS and CPT sensitivity (assays of DSB repair) while retaining sufficient Tel1 activity to carry out some Rad53 phosphorylation and maintain telomeres better than rad50 null cells (Figures 2A, 3B,C). Who is to say that it is partial loss of Tel1 activity, and not the partial loss of DSB repair activity, that is more critical for tumor suppression? Perhaps both are important. The same criticisms apply to the other supposed strong SOF mutations.

An obvious way to address these deficiencies is carry out genetic epistasis studies. For example, if rad50-L1240F actually is a “SOF mutation that blocks Tel1/ATM activation while leaving DSB repair largely unaffected”, one would expect to see no genetic synergism between rad50-L1240F and tel1 null mutations. Actually, as part of an experiment to address another question, a rad50-L1240F tel1 null strain is found to be much more sensitive to CPT than tel1 null cells (Figure 5B). By the cell dilution estimation method that difference appears to be at least 3,000-fold. Furthermore, because the rad50-L1240F tel1 null strain appears to be much more sensitive to CPT than rad50-L1240F (comparing 20 micromolar CPT data in Figures 2A and 5B indicates a >3,000-fold difference), then rad50-L1240F cells must retain substantial Tel1 activity. Faced with these data, it is difficult to conclude that rad50-L1240F is a useful SOF mutation.

The same criticisms apply to the other supposed strong SOF mutations; for example, the mre11-E38K and tel1 null mutations appear to be strongly synergistic in the CPT survival assays (Figures 2A and 5B).

It seems that more careful and extensive tel1 null epistasis analyses would be a huge help to testing the conclusions of this study. The authors would know best, but at least tests with MMS and including additional controls. A C-terminal truncation of Xrs2, which eliminates Tel1 interactions but leaves MRX repair activities intact (a real strong SOF) could be a very useful control. Genetic epistasis studies with mutants defective in alternative resection pathways, i.e. exo1 null and sgs1 null, could also be critically informative. If the MRX resection activity is largely unaffected by putative SOF alleles of rad50 or mre11, one would not expect genetic synergisms with exo1 null or sgs1 null.

Also, please check references. A citation for tumor mutations yields a yeast paper (16).

Reviewer #2: Hohl et al. screened cancer derived Rad50 and Mre11 mutations through clinical sequencing program at SKCC and those recurrent and conserved mutations were analyzed for their DNA repair and DDR functions using various genetic and biochemical assays in budding yeast. Most of these mutation attenuate ATM activity while they still sustain DNA repair functions. The authors argue that inactivation of ATM-MRE11 axis confers advantages to tumor cells whereas DNA repair defects impede carcinogenesis.

Analyzing the tumor borne Rad50 and Mre11 mutations in 5 % among cancers offer unique opportunity to interrogate the function of the Mre11/Rad50 complex in tumor suppression and the key attributes from its diverse functions pertinent for tumorigenesis. Yeast provides an excellent vehicle to investigate its biochemical and genetic deficits using a range of genetic and biochemical assays. Such systematic and comprehensive analyses help deduce useful insights into the cancer driving defects in cancer etiology. The assays employed by the authors are extensive: drug sensitivity, the integrity of NHEJ and meiotic recombination, telomere length, Tel1 activity, ATP binding/hydrolysis and electro-mobility shift assay, DSB association using ChIP assays. Together, the authors propose that the cancer associated Rad50 and Mre11 mutation alleles are separation of function mutations deficient in ATM activation but largely sustaining DNA repair function. Using molecular modeling, the authors propose that one of the rad50 (D67Y/N) mutations are defective in the ADP release, raising the possibility if the transition of Mre11/Rad50 from an open to close state triggers ATM activation.

Despite these positives and the unique strategies opted to define tumor relevant functions of Mre11/Rad50 complex, additional points need to be addressed and modified accordingly in the manuscript.

1. It appears that the underlying defects in Mre11/Rad50 mutations cannot be simply explained by ATM activation defect. In fact, the phenotypes associated with the cancer mutations are distinct from those in tel1-kd or tel1 deletion mutants. It should be useful to include Tel1 deletion- or kd mutants in the assays and compare them with those in rad50 and mre11 mutants. It should be useful to comment these points in the manuscript. One wonder if partial and not full inactivation of ATM is more tumorigenic as ATM activity is also critical for HR due to its role in resection.

2. Curiously missing among the analysis is the integrity of HR in rad50 and mre11 mutants. It should be important to analyze the integrity of allelic and/or ectopic recombination after HO induced DSB formation and the types of repair events. It should be also helpful if the mutants support resection function using resection assay after HO expression in yeast cells lacking homologous donors.

3. Mre11 complex are important for replication progression and integrity under DNA damaging condition. This is particularly important to test as it is likely relevant to tumor suppression. One can assess these activities roughly by assessing the doubling time of the mutant yeast. The authors should check replisome stability and GCR rate as shown in Oh et al (Cell Rep. 2018 Nov 13;25(7):1681-1692).

4. The key question is why ATM defect benefit tumorigenesis. ATM is also prerequisite to ATR activation as CtIP phosphorylation by ATM is essential for resection. ATR deficiency could accelerate replication stress and drive mutagenesis. The author should show if Rad53 activity with or without DNA damage is aberrant in Mec1 proficient cells.

5. It is confusing that the authors use both yeast and human designation of mutations in the text. It should be easier for the readers if they used only yeast or human numbers throughout and fully disclose if other numbers are used for clarity.

6. In page 8, line 4, L1240F should be 1259C.

7. If L1240F and D67N/Y mutations disrupt the Tel1 activity by two different means, can it be validated to test if the double mutant is more defective in Tel1 activity?

8. It is puzzling as L1240F is deficient in DNA binding according to EMSA (Fig. 5C) but shows increased Xrs2-HA and Tel1 association at DNA break according to ChIP (Fig. 5A). This needs some explanation.

Reviewer #3: This ms. describes an important approach to defining what aspect of a gene, complex or pathway is affected in tumor cells by examining the phenotype of recurrent mutations in a model system. The effort here includes DNA sequence analysis of more than 40,000 tumor samples to search for mutations in the human MRN complex. Importantly, mutations in this complex are found in ~5% of all solid tumors. Mutations in residues conserved between the human and yeast Mre11 and Rad50 proteins were chosen for modeling in yeast, where the Petrini lab and others have done extensive phenotypic analysis. Although the screen was biased for mutations in conserved residues, it is striking that many of them define separation of function alleles between the DNA damage response signaling and direct effects on genetic recombination (broadly defined as clastogen sensitivity, spore viability and NHEJ proficiency). Importantly, 40% of the modeled mutations affect DNA damage signaling. There is also an excursion into a reversion analysis of one allele, rad50-L1240F, which although interesting on its own, for the amount of effort, does not really add much to the main point of the ms. Next the molecular basis of the phenotype of many of the alleles was further investigated by a plethora of assays, including their effects on Xrs2 and Tel1 recruitment to an HO-induced DSB at the MAT locus. This data led them to speculate on which mutations affect the MRX-Tel1 interaction. Next they examined DNA binding, ATP binding, ATP hydrolysis and Tel1 mediated phosphorylation of several of the mutants. Finally, they modeled the yeast MR complex as a short coiled coil using structural data from other eukaryotes. They speculate from these comparisons that the loss of function of one class of mutations is due to impaired release of ADP and an altered transition between the open and closed state, rather than one state or the other.

As seen from the summary, this ms. contains a huge amount of data that centers around mutations found in tumor samples that inactivate the MRN complex. The main point of the ms. is that the tumor suppressor activity of the MRN complex resides in its activation of ATM signaling and that its loss is important during tumorigenesis. Taken at face value, this view is well supported by their data. However, the ms. suffers from a “flow” problem in that there is a huge amount of data that is presented to make the this aforementioned point and the reader gets lost as it relates to one overarching view. One specific example is the comparison of rad50-D67 and rad50-L1240 mutations, which are nearby in the structural model, but exhibit very different behavior with respect to proteins levels, DNA binding and Tel1 recruitment.

While reading the list of mutations that behave this way or that way, one loses focus on what aspect of each phenotype is important for which behavior of the complex. My recommendation is that the more attention be paid to tracing the path through each of these studies in a more precise way - perhaps some data can be relegated to the supplemental material an the reader be directed toward the overarching goal of the ms., the function of MRN in tumor suppression. I realize that the suppressor study was a large amount of work, but the payoff is small compared to the effort. The same can be said for the molecular characterization - there is a lot of data, but in the end, one gets lost in the recounting of the individual phenotypic result.

Finally, the molecular modeling at the end of the ms. is striking, but it seems to have been tacked on to an already pretty large story. Was any of this molecular information used to guide any of the mutant modeling that was done in yeast? Should it be part of a bigger biochemical study?

In the end, the expectations at the beginning of this ms. are high and are almost achieved. It would be helpful to focus the ms. on those points that directly relate to the thesis at hand.

A few other points:

Pg. 5, bottom: It would be useful to have some discussion of when these mutations may arise during tumorigeneis. Have the authors looked at tumors in various stages of the cancer?

Do the Rad53 phosphorylation data in Figures 3 and 4 really show anything helpful? The increases that are pointed to seem non-existent to this reader.

Pg. 20, last sentence of the 2nd ¶: It is unclear how the CPT sensitivity of the mre11-E38K relates to Tel1 binding to the MRX complex.

Minor details:

Pg. 20, ln. 6: This suggests… -> This “what” suggests? These observations suggest…? (from both of the preceding sentences) or This reduction of the ChIP signal suggests…? (from the last of the two sentences)

In general, “phenotype” is singular and is often used to describe the collection of aspects of a genotype. Thus, a mutation’s effect on sporulation, growth and checkpoint activation are three different aspects of its phenotype (singular). When more than one mutation is being compared, then “phenotypes” can be used to lump together the genotypes. Thus, the phenotypes of the three alleles are the same with respect to sporulation.

Legend for Table S2: Rewrite “Related to Fig 1A. This is an extended version of above the table shown in Fig 1A,…” -> Related to Fig 1A and is an extended version of the table shown, listing all allele frequencies, number of mutations present in tumor samples and sequencing database source.

Supporting Table S2, line 4: Change “homozygote” to “a homozygous” (homozygote just seems a little awkward)

It’s not a huge issue, but MATa should be MAT”a- bold, not italics” and MATalpha should be MAT”not bold and not italics, alpha symbol”

**Have all data underlying the figures and results presented in the manuscript been provided?**

Reviewer #1: Yes

Reviewer #2: Yes

Reviewer #3: Yes

PLOS authors have the option to publish the peer review history of their article (what does this mean?). If published, this will include your full peer review and any attached files.

Reviewer #1: No

Reviewer #2: No

Reviewer #3: No

---

## [Decision Letter · Decision Letter 1]

7 Jan 2020

Dear Dr Petrini,

Thank you very much for submitting your Research Article entitled 'Modeling Cancer Genomic Data in Yeast Reveals Selection Against ATM Function During Tumorigenesis' to PLOS Genetics. Your manuscript was fully evaluated at the editorial level and by independent peer reviewers. The reviewers appreciated the attention to an important topic but identified some aspects of the manuscript that should be improved.

We therefore ask you to modify the manuscript according to the review recommendations before we can consider your manuscript for acceptance. Your revisions should address the specific points made by each reviewer. Reviewer 1 is still concerned that the separation of function phenotype is overstated because the rad50-L1240F and mre11-E38K mutants are not WT for DNA repair functions, and mre11-E38K synergizes with tel1-kd, in contrast to what is stated in the text. Sporulation and spore viability of the rad50L1240F mutant seem quite a bit lower than WT. Please can you add p values to Figs 2D. I find it quite surprising that telomere length is not more strongly affected by the mutations, suggesting that they are not completely compromised for Tel1 activation. I think part of the problem with making general conclusions about the mutations is that each behaves a little differently in the various assays described, reflecting the complex functions of the MRX/N in DNA repair and signaling. 

[LINK]

Yours sincerely,

Lorraine S. Symington

Associate Editor

PLOS Genetics

Gregory P. Copenhaver

Editor-in-Chief

PLOS Genetics

Reviewer's Responses to Questions

**Comments to the Authors:**

Reviewer #1: My previous primary criticism, which was shared with another reviewer, was that key rad50/mre11 alleles reported in this paper did not behave as straightforward SOF mutations that ablate MRX-dependent Tel1 function without impairing MRX DNA repair functions. I noted that their effects in DNA repair assays (e.g. CPT sensitivity) were strongly additive with a tel1 deletion mutation. The authors reply that epistasis tests may be misinformative with these alleles because they are not simple null mutations. They propose that the alleles in question destabilize the MRX complex to make its DNA repair function dependent on a physical interaction with Tel1. The principal reason for proposing this model is that they believe the alleles show little or no synergism with a tel1KD (kinase dead) alleles. In point C of “supporting this concept”, the rebuttal states: “tel1KD rad50-L1240F or tel1KD mre11-E38K double mutants are as CPT or MMS resistant as either single tel1KD or rad50-L1240F, mre11-E38K and xrs2-664 (the Xrs2 C terminal truncation [1] mutant invoked by Reviewer #2).” However, to my eyes the data in Figure 4 contradict this statement. From the 5-fold serial dilutions on 25 micromolar CPT, tel1KD rad50-L1240F appears to be about 5-fold more sensitive than tel1KD and >25-fold more sensitive than rad50-L1240F. Similar differences are obvious for 0.020 micromolar MMS and 200 mM HU. Likewise, tel1KD mre11-E38K is clearly much more sensitive to CPT, MMS and HU than either tel1KD or mre11-E38K single mutants (Figure 4). (Side note: I cannot find any data for xrs2-664 in the manuscript, so it is unclear how the authors know how it compares to tel1KD rad50-L1240F or tel1KD mre11-E38K).

The rebuttal further states: “We consider these two alleles to be SOF since they are significantly more defective in Tel1 activation by several criteria than they are in DSB repair”. I would not feel confident in drawing this conclusion from these data. There is no objective or unambiguous way to compare the assays. If I were to propose that the rad50-L1240F or mre11-E38K mutations have reduced MRX-dependent Tel1 activation and DNA repair activities both by about 75%, I cannot identify any data in this manuscript that would convincingly disprove this hypothesis.

Reviewer #3: The ms. has been improved significantly and is much easier to read. There is still some "roughness" in the presentation of the molecular phenotype and biochemical data concerning the SOF mutations. Given the concerns of two of the reviewers, this section needs a little more attention to make it clear what can and cannot be said about the SOF phenotype of each allele. The real take-home message is that MRN mutations play a complex tumor suppressor role. Indeed it is quite clear that the recombination function is likely not important for this role. It might be worthwhile to emphasize this result in the title and indicate that DNA repair functions of the MRN complex are not necessary for tumor suppression.

Typos:

pg. 18, ln. …as well biochemical… -> …as well as biochemical…

pg. 19, ln. 12: …activation defects Rad50-1240F… -> …activation defects in (or of) Rad50-1240F…

pg. 14, ln. 11; pg. 19, last ln.; pg. 20, ln. 3; pg. 47 last ln.: In each case, "This 'what?'”

pg. 21, 6th ln. from bottom: “)” has no partner “(“

**Have all data underlying the figures and results presented in the manuscript been provided?**

Reviewer #1: Yes

Reviewer #3: None

PLOS authors have the option to publish the peer review history of their article (what does this mean?). If published, this will include your full peer review and any attached files.

Reviewer #1: No

Reviewer #3: No

---

## [Editor Report · Decision Letter 2]

19 Jan 2020

Dear Dr Petrini,

We are pleased to inform you that your manuscript entitled "Modeling Cancer Genomic Data in Yeast Reveals Selection Against ATM Function During Tumorigenesis" has been editorially accepted for publication in PLOS Genetics. Congratulations!

Yours sincerely,

Lorraine S. Symington

Associate Editor

PLOS Genetics

Gregory P. Copenhaver

Editor-in-Chief

PLOS Genetics

Comments from the reviewers (if applicable):

**Data Deposition**

http://datadryad.org/submit?journalID=pgenetics&manu=PGENETICS-D-19-01515R2

**Press Queries**

---

## [Editor Report · Acceptance letter]

6 Mar 2020

PGENETICS-D-19-01515R2 

Modeling Cancer Genomic Data in Yeast Reveals Selection Against ATM Function During Tumorigenesis 

Dear Dr Petrini, 

We are pleased to inform you that your manuscript entitled "Modeling Cancer Genomic Data in Yeast Reveals Selection Against ATM Function During Tumorigenesis" has been formally accepted for publication in PLOS Genetics! Your manuscript is now with our production department and you will be notified of the publication date in due course.

With kind regards,

Matt Lyles

PLOS Genetics

On behalf of:
